# In vitro generation of human pluripotent stem cell derived lung organoids

**Briana R Dye[1], David R Hill[2], Michael AH Ferguson[2], Yu-Hwai Tsai[2], Melinda S Nagy[2], Rachel Dyal[2], James M Wells[3,4], Christopher N Mayhew[3], Roy Nattiv[5], Ophir D Klein[5,6,7], Eric S White[2], Gail H Deutsch[8], Jason R Spence[1,2,9]\***

[1]Department of Cell and Developmental Biology, University of Michigan Medical School, Ann Arbor, United States; [2]Department of Internal Medicine, University of Michigan Medical School, Ann Arbor, United States; [3]Division of Developmental Biology, Cincinnati Children's Hospital Medical Center, Cincinnati, United States; [4]Division of Endocrinology, Cincinnati Children's Hospital Medical Center, Cincinnati, United States; [5]Institute for Human Genetics, Department of Pediatrics, University of California, San Francisco, San Francisco, United States; [6]Program in Craniofacial and Mesenchymal Biology, University of California, San Francisco, San Francisco, United States; [7]Center for Craniofacial Anomalies, University of California, San Francisco, San Francisco, United States; [8]Department of Laboratories, Seattle Children's Hospital and University of Washington, Seattle, United States; [9]Center for Organogenesis, University of Michigan Medical School, Ann Arbor, United States

**Abstract** Recent breakthroughs in 3-dimensional (3D) organoid cultures for many organ systems have led to new physiologically complex in vitro models to study human development and disease. Here, we report the step-wise differentiation of human pluripotent stem cells (hPSCs) (embryonic and induced) into lung organoids. By manipulating developmental signaling pathways hPSCs generate ventral-anterior foregut spheroids, which are then expanded into human lung organoids (HLOs). HLOs consist of epithelial and mesenchymal compartments of the lung, organized with structural features similar to the native lung. HLOs possess upper airway-like epithelium with basal cells and immature ciliated cells surrounded by smooth muscle and myofibroblasts as well as an alveolar-like domain with appropriate cell types. Using RNA-sequencing, we show that HLOs are remarkably similar to human fetal lung based on global transcriptional profiles, suggesting that HLOs are an excellent model to study human lung development, maturation and disease.

**\*For correspondence:** spencejr@ umich.edu

**Competing interests:** The authors declare that no competing interests exist.

## Introduction

Several reports have demonstrated that directed differentiation of human pluripotent stem cells (hPSCs), which include embryonic (hESCs) and induced (iPSCs) stem cells, is one of the most efficient approaches to achieving differentiation of a cell or tissue of interest (*D'Amour et al., 2005*; *Kroon et al., 2008*; *Si-Tayeb et al., 2009*; *Spence et al., 2011*; *Wong et al., 2012*). Using this approach, differentiation of hPSCs into lung lineages has been achieved using diverse methodology with varying degrees of success (*Kadzik and Morrisey, 2012*; *Longmire et al., 2012*; *Mou et al., 2012*; *Wong et al., 2012*; *Ghaedi et al., 2013*; *Huang et al., 2013*; *Firth et al., 2014*).

Thus far, the majority of efforts to differentiate lung lineages from hPSCs have focused on using 2-dimensional (2D) monolayer cultures. Several recent advances in generating 3-dimensional (3D) organ-like tissues, called 'organoids', have been reported (*Meyer et al., 2011*; *Spence et al., 2011*; *Nakano et al., 2012*; *Takebe et al., 2013*; *Lancaster et al., 2013*; *McCracken et al., 2014*). Such 3D

**eLife digest** Cell behavior has traditionally been studied in the lab in two-dimensional situations, where cells are grown in thin layers on cell-culture dishes. However, most cells in the body exist in a three-dimensional environment as part of complex tissues and organs, and so researchers have been attempting to re-create these environments in the lab. To date, several such 'organoids' have been successfully generated, including models of the human intestine, stomach, brain and liver. These organoids can mimic the responses of real tissues and can be used to investigate how organs form, change with disease, and how they might respond to potential therapies.

Here, Dye et al. developed a new three-dimensional model of the human lung by coaxing human stem cells to become specific types of cells that then formed complex tissues in a petri dish. To make these lung organoids, Dye et al. manipulated several of the signaling pathways that control the formation of organs during the development of animal embryos. First, the stem cells were instructed to form a type of tissue called endoderm, which is found in early embryos and gives rise to the lung, liver and other several other internal organs.

Then, Dye et al. activated two important developmental pathways that are known to make endoderm form three-dimensional intestinal tissue. However, by inhibiting two other key developmental pathways at the same time, the endoderm became tissue that resembles the early lung found in embryos instead.

This early lung-like tissue formed three-dimensional spherical structures as it developed. The next challenge was to make these structures develop into lung tissue. Dye et al. worked out a method to do this, which involved exposing the cells to additional proteins that are involved in lung development. The resulting lung organoids survived in laboratory cultures for over 100 days and developed into well-organized structures that contain many of the types of cells found in the lung.

Further analysis revealed the gene activity in the lung organoids resembles that of the lung of a developing human fetus, suggesting that lung organoids grown in the dish are not fully mature. Dye et al.'s findings provide a new approach for creating human lung organoids in culture that may open up new avenues for investigating lung development and diseases.

models offer several advantages; they often possess structural organization similar to the native organ, cell types from multiple germ layers (for example, mesoderm and endoderm (*Spence et al., 2011*; *McCracken et al., 2014*; *Wells and Spence, 2014*), and multiple cellular lineages, making them a physiologically complex model to study developmental processes, tissue homeostasis and pathological conditions in vitro.

Previous work has demonstrated that activation of FGF and WNT signaling synergistically drives CDX2+ intestinal lineage commitment in hPSC-derived endoderm and also drives 'morphogenesis in a dish', where the 2D tissues self-organize into 3D spheroids comprised of mesenchymal and polarized epithelial layers that detach from the adherent cell layer (*Spence et al., 2011*). It has also been demonstrated that inhibition of BMP and TGFβ signaling is able to drive tissue into a SOX2+ foregut lineage (*Green et al., 2011*; *McCracken et al., 2014*). Building on these previous studies, we show that simultaneous stimulation of WNT and FGF signaling while inhibiting BMP/TGFβ signaling pathways in hPSC-derived endoderm cultures prevents intestinal lineage commitment, and instead, favors a SOX2+ anterior foregut fate while also robustly generating SOX2+ anterior foregut 3D spheroid structures.

In order to further restrict foregut spheroids to the lung lineage, the current study focused on manipulating FGF and HH signaling. In the mouse, high levels of Fgf signaling have been shown to induce Shh expression in the lung endoderm (*Hebrok et al., 1998*; *Morrisey and Hogan, 2010*; *Rankin and Zorn, 2014*) which is accompanied by induction of the Nkx2.1+ lung progenitor field (*Hebrok et al., 1998*; *Serls, 2004*). Shh then signals from the endoderm to the mesoderm, and mutations in *Shh*, *Gli2* or *Gli3* lead to perturbed lung development, with *Gli2/Gli3* double knockout mice showing lung agenesis (*Bellusci et al., 1997a*; *Motoyama et al., 1998*; *Li et al., 2004*). Our results demonstrate that FGF2 induces NKX2.1, PAX8, and SHH in human foregut endoderm cultures. By using pharmacological inhibitors of FGF and HH signaling we demonstrate that SHH is required for NKX2.1 expression downstream of FGF2, and that FGF2 also induces PAX8 independently of HH

signaling. These observations suggest a paradigm where FGF$^{Lo}$/HH$^{Hi}$ conditions preferentially induce PAX8$^{Lo}$/NKX2.1$^{Hi}$ lung progenitors and FGF$^{Hi}$/HH$^{Lo}$ conditions favor a PAX8$^{Hi}$/NKX2.1$^{Lo}$ fate. Given that Pax8 is required for thyroid development, we focused on defining the most robust conditions to induce NKX2.1 while minimizing PAX8 expression (*Kimura et al., 1996*; *Mansouri et al., 1998*; *Yuan et al., 2000*; *Vilain et al., 2001*; *Li et al., 2004*; *Kusakabe et al., 2006*; *Carré et al., 2009*; *Narumi et al., 2012*). By applying HH$^{Hi}$ conditions during generation of foregut spheroids we were able to enhance NKX2.1 expression in foregut spheroids and subsequently expand spheroids in media containing FGF10, allowing them to grow into organoids. Organoids persisted in culture for over 100 days and developed well-organized proximal-like airway epithelial structures that included many cell types found in the proximal lung epithelium, including basal and ciliated cells along with rare club cells. Moreover, proximal airway structures were often surrounded by smooth muscle actin (SMA) positive mesenchymal tissue. Organoids also possessed distal-like epithelial cells that co-expressed progenitor markers, SFTPC/SOX9 and HOPX/SOX9, consistent with early bipotent alveolar progenitor cells seen in mice (*Desai et al., 2014*; *Treutlein et al., 2014*). To support the idea that organoids may be more similar to a developing lung with abundant progenitor cells, we used RNA-sequencing to compare the global transcriptional profile of organoids to the human fetal and adult lung, undifferentiated hESCs and definitive endoderm. Principal component analysis, hierarchical clustering and Spearman's correlation all show that organoids have striking similarity to the human fetal lung.

Taken together, our data demonstrates an efficient and robust in vitro system to generate complex, 3D human lung organoids that are immature/fetal in nature. We anticipate that this model will serve as an unparalleled model for the study of human lung development, maturation and disease.

## Results

### Differentiation of hPSCs into anterior foregut spheroids

We and others have reported efficient induction of human endoderm using ActivinA (*D'Amour et al., 2005*; *Zhang et al., 2010*; *Spence et al., 2011*), and a further lineage restriction into SOX2+ anterior foregut endoderm using inhibition of BMP and TGFβ signaling (*Green et al., 2011*; *Loh et al., 2014*). We have recently demonstrated that inhibition of BMP signaling during intestinal lineage induction with WNT and FGF ligands is sufficient to inhibit intestinal CDX2 and induce SOX2+ posterior foregut spheroids capable of giving rise to human gastric (antral) organoids (*McCracken et al., 2014*). Given that the lung is derived from the anterior foregut, we sought to define conditions to generate ventral anterior foregut spheroids. To do this, we tested if dual inhibition of BMP and TGFβ was able to anteriorize cultures, as previously described (*Green et al., 2011*). We treated hESCs with ActivinA (100 ng/ml) for 4 days to induce endoderm, followed by 4 days of Noggin (NOG, 200 ng/ml) and the small molecule TGFβ inhibitor, SB431542 (SB, 10 μM). We confirmed that these conditions were able to induce robust mRNA and protein expression of SOX2, which co-expressed the endodermal marker FOXA2, while repressing the intestinal lineage marker CDX2 (*Figure 1A–C*, *Figure 1—figure supplement 1A*). QRT-PCR analysis also showed that compared to controls (in which endoderm was induced but was not exposed to NOG/SB), exposure to NOG/SB robustly induced ventral anterior foregut genes *NKX2.1* and *PAX8*, while the posterior foregut transcript, *PDX1* was reduced. *HHEX*, which is expressed in the developing liver, biliary system and thyroid, but is absent from the lung primordium, remained unchanged (*Figure 1B*). Given that NKX2.1 is expressed in the lung and thyroid primordium, and PAX8 is expressed in the thyroid primordium, these results suggest that 4 day ActivinA treatment followed by a 4 day NOG/SB treatment biases the cultures towards ventral-anterior foregut lineages.

Addition of FGF4 plus WNT3A (or Chir99021, a GSK3β inhibitor that enhances β-catenin dependent WNT signaling) promotes CDX2 intestinal lineage commitment and 3D spheroid formation in endoderm cultures (*Spence et al., 2011*; *Xue et al., 2013*; *Chen et al., 2014b*). Based on our results in *Figure 1B–C*, we hypothesized that combining FGF, Chir99021, NOG and SB would result in the generation of SOX2+ ventral-anterior foregut spheroids. To test this, we generated endoderm (4 days ACTA) and added no growth factors (Endoderm controls) or NOG, SB, FGF4, and Chir99021 (NOG/SB/F/Ch) (*Figure 1D*). Addition of all four factors resulted in the generation of 3-dimensional SOX2+, CDX2− spheroids (*Figure 1E,F*). SOX2+ spheroids also expressed the endodermal protein FOXA2, and were epithelial, co-expressing E-Cadherin (ECAD) (*Figure 1F*,

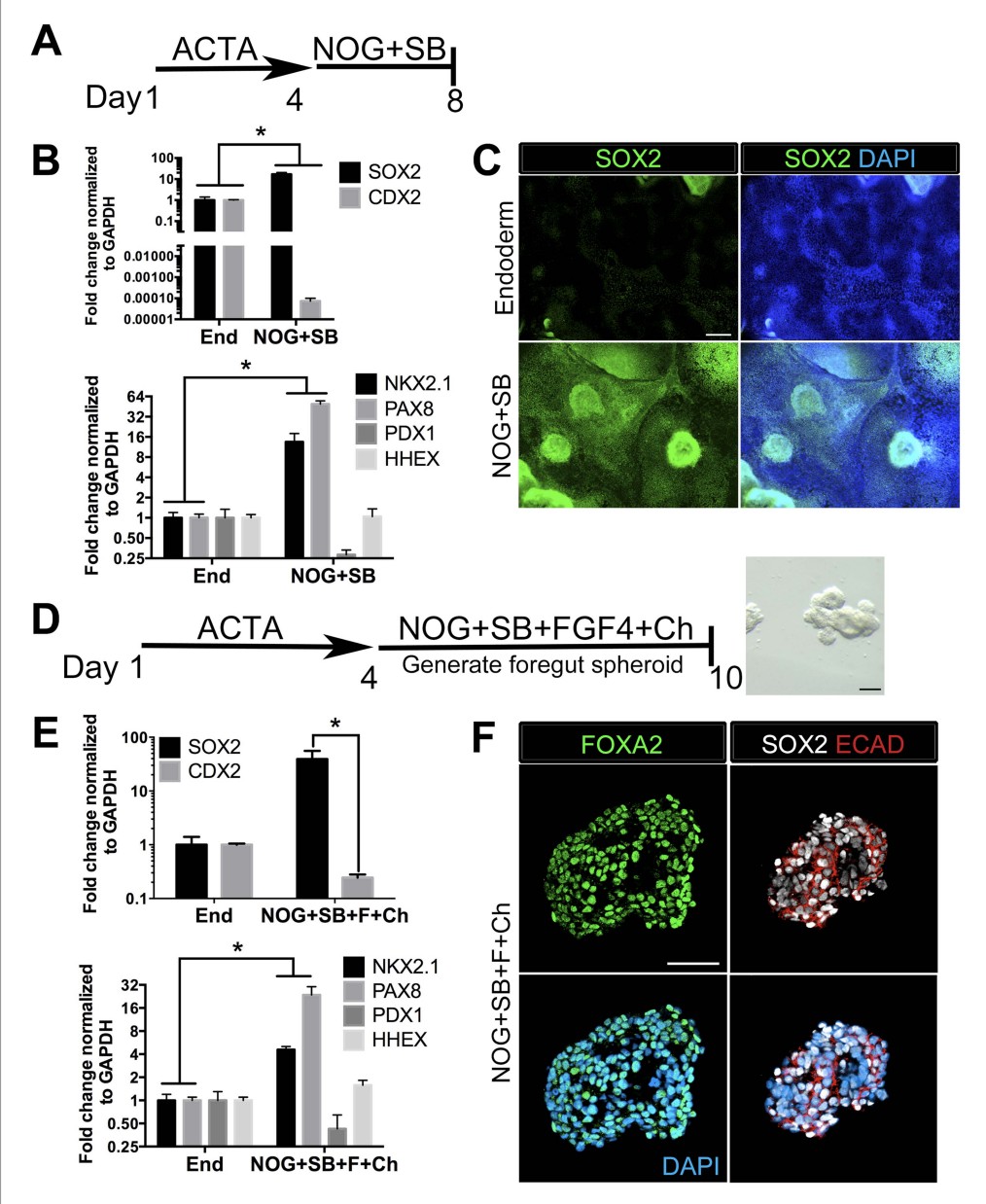

**Figure 1**. Generation of three-dimensional ventral anterior foregut spheroids from endoderm monolayers. (**A**) hESCs were differentiated into foregut endoderm by treating cells with 4 days of Activin A (ACTA) followed by 4 days of NOG+SB. (**B**) Foregut endoderm (NOG+SB) had high expression of the foregut marker *SOX2* while the hindgut marker *CDX2* was significantly reduced compared to untreated endoderm controls (End). NOG+SB monolayers had high expression of ventral anterior foregut genes *NKX2.1* and *PAX8* while the posterior foregut marker *PDX1* was reduced. The foregut marker *HHEX* is expressed in the developing liver, biliary system, and thyroid and remained unchanged. (**C**) The majority of cells in NOG+SB treated cultures were SOX2 positive (green) compared to the control, in which only scattered clusters of cells were SOX2 positive. The scale bar represents 200 μm. (**D**) hESCs were differentiated into foregut spheroids by treating cells with 4 days of ACTA and then additional 4–6 days of NOG+SB+FGF4+Ch. Representative images of a spheroid in a matrigel droplet are shown as a whole mount image. Scale bar represents 100 μm. (**E**) Foregut spheroids (NOG+SB+FGF4+Ch) had high expression of the foregut marker *SOX2* while the hindgut marker *CDX2* was significantly reduced compared to untreated endoderm control (End) (top panel). Spheroids had high expression of anterior foregut genes *NKX2.1* and *PAX8* while the posterior foregut marker *PDX1* was

*Figure 1. continued on next page*

*Figure 1. Continued*

reduced and HHEX was unchanged (bottom panel). *p < 0.05, error bars represent SEM. (**F**) The majority of cells in foregut spheroids are FOXA2+ (green, left panel) and SOX2+ (white, right panel) and ECAD+ (red, right panel). Scale bar represent 50 μm.

The following figure supplements are available for figure 1:

**Figure supplement 1**. Monolayer cultures express lung specific markers.

**Figure supplement 2**. Foregut spheroids co-express endoderm and lung specific markers.

**Figure supplement 3**. Foregut spheroids consist of both epithelial and mesenchymal cells.

**Figure supplement 4**. NOG+SB+FGF4+Ch spheroids do not express neural markers.

*Figure 1—figure supplement 2*). In addition to SOX2, spheroids exhibited higher mRNA expression of anterior foregut lineage markers *NKX2.1* and *PAX8* compared to endoderm controls, suggesting that they are ventral-anterior foregut spheroids (*Figure 1E*), however, immunofluorescence revealed that levels of NKX2.1 protein were just above the detection threshold (*Figure 1—figure supplement 2*). Spheroids also possess a minor population of cells that are mesodermal in origin staining positive for Vimentin protein (VIM) (*Figure 1—figure supplement 3*). Given that neural tissues also express NKX2.1, PAX8, SOX2, and FOXA2, and that neural induction protocols use dual BMP and TGFβ inhibition, we wanted to exclude the possibility that spheroids were neural in nature. To do this, we generated endoderm control cultures, foregut spheroids (ActivinA followed by NOG/SB/F/Ch), and induced neural tissue by adding NOG/SB to hESC cultures that were not treated with ActivinA (*Chambers et al., 2009*). By examining induction of the neural markers *NESTIN, SOX1,* and *PAX6*, we confirmed that these transcripts were highly induced in dual NOG/SB neural cultures, but were low in ventral foregut spheroid cultures. In contrast, *FOXA2*, which is expressed in the foregut (*Monaghan et al., 1993*; *Ang and Rossant, 1994*; *D'Amour et al., 2005, 2006*; *Kroon et al., 2008*; *Si-Tayeb et al., 2009*; *DeLaForest et al., 2011*) and in some neural tissues (*Stott et al., 2013*), had high expression in ventral foregut spheroids, but was significantly reduced in dual NOG/SB neural conditions (*Figure 1—figure supplement 4*). Taken together, these results strongly suggest spheroids are indeed foregut, and not of neural origin.

## Induction of anterior foregut endoderm into a lung lineage through modulation of FGF and HH signaling

Many signaling pathways are important for lung induction and development (reviewed in *Min et al., 1998*; *Weaver et al., 2000*; *Morrisey and Hogan, 2010*; *Rankin and Zorn, 2014*). High levels of Fgf signaling have been shown to induce Shh and Nkx2.1 expression in the foregut endoderm in mice (*Hebrok et al., 1998*; *Serls, 2004*); furthermore, Gli2/3 null mouse embryos fail to form lungs (*Motoyama et al., 1998*) and Hh signaling is important for lung mesenchyme proliferation in vivo (*Bellusci et al., 1997a*). These data confirm that Fgf and Hh signaling are critical for lung specification and ligands from both signaling pathways have been applied to hPSC derived lung lineages in 2D cultures (*Wong et al., 2012*; *Huang et al., 2013*). In our cultures we have reported that approximately 85–95% of cells are endoderm, but a portion of the remaining cells are mesodermal and this small mesodermal population is maintained in the spheroids and organoids (*Spence et al., 2011*; *McCracken et al., 2014*) (*Figure 1—figure supplement 3*). Therefore, based on mouse and hPSC studies, we hypothesized that FGF and/or HH signaling would induce an NKX2.1+ lung lineage in anterior foregut endoderm. To test our hypothesis we initially focused on adherent endoderm monolayer cultures to optimize induction conditions. Cultures were treated for 4 days with ActivinA followed by an additional 4 days with NOG/SB (referred to as Foregut). Controls consisted of ActivinA treatment only followed by no additional growth factors (Endoderm controls), or ActivinA followed by NOG/SB, followed by no additional factors (Foregut controls). All experimental groups were compared to both endoderm and foregut controls (*Figure 2*). We first tested the ability of FGF2 to

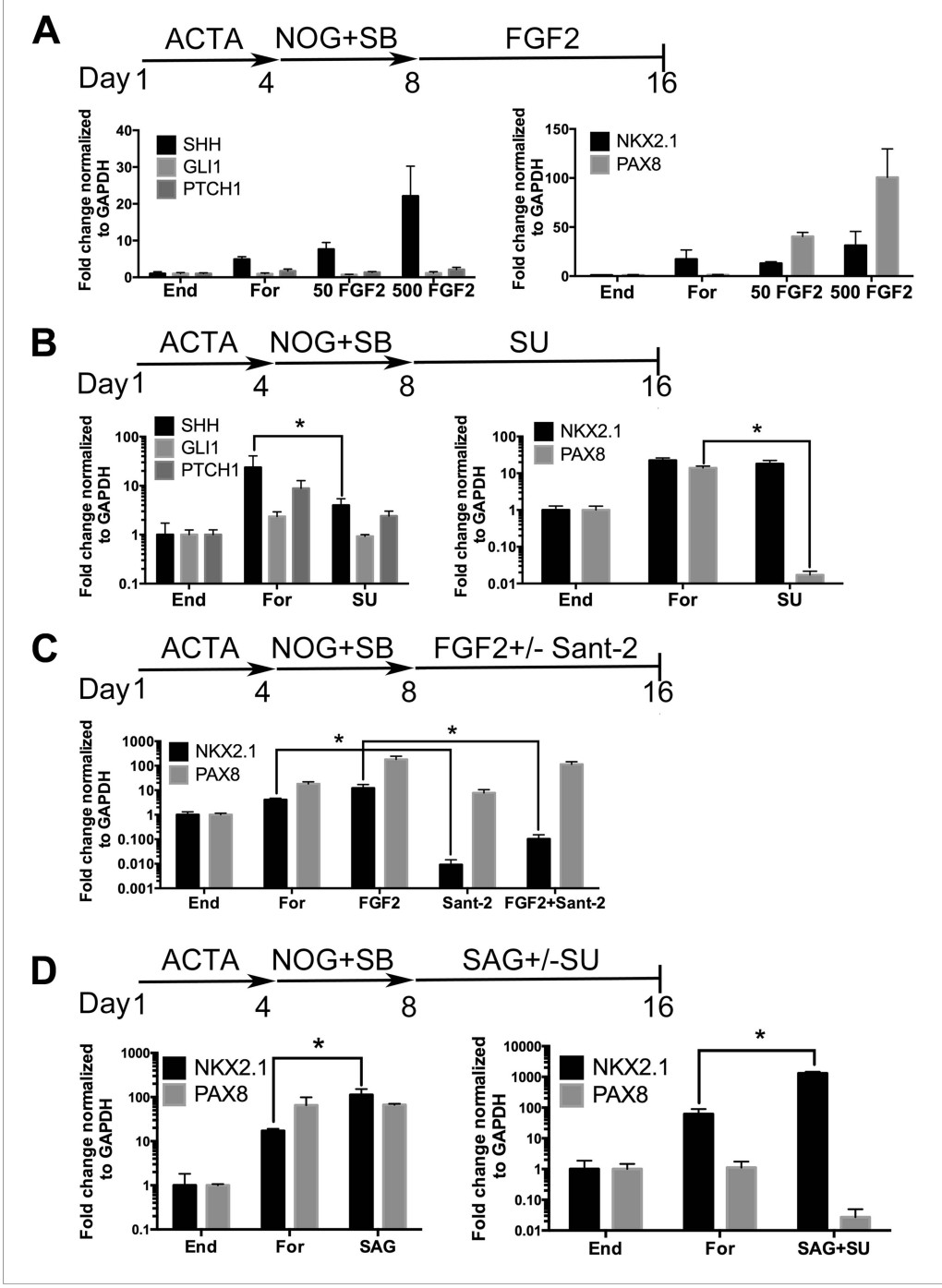

**Figure 2**. Induction of NKX2.1 in anterior foregut endoderm by modulating FGF and HH signaling. (**A**) hESCs were differentiated into endoderm (End) or anterior foregut with NOG+SB (For). Anterior foregut was treated with low (50 ng/ml) and high (500 ng/ml) concentrations of FGF2. FGF2 caused a dose-dependent increase in *SHH* and *PAX8* expression with a modest increase in *NKX2.1* expression compared to untreated endoderm controls. Note that *NKX2.1* expression is increased by NOG+SB exposure alone (no FGF2). (**B**) Addition of the FGF inhibitor SU5402 (SU) to NOG+SB foregut cultures (For) caused a significant reduction of *SHH* and *PAX8* expression, but *NKX2.1*, *GLI1*, and *PTCH1* were not significantly different compared to the foregut controls, in which no growth factors were added after SB+NOG. (**C**) Addition of the HH inhibitor Sant-2 caused a significant reduction in *NKX2.1* compared to foregut control. Similarly when FGF2 (500 ng/ml) and Sant-2 were added simultaneously, the modest *NKX2.1* induction caused by FGF2 was significantly reduced whereas *PAX8* expression remained unchanged. (**D**) Foregut endoderm treated with SAG or SAG+SU for 8 days had a 6.5-fold and 21-fold increase of *NKX2.1* expression,
*Figure 2. continued on next page*

*Figure 2. Continued*

respectively, compared to untreated foregut controls. *PAX8* expression was unchanged in the SAG treated cultures whereas SAG+SU treated cultures demonstrated a 41-fold decrease in *PAX8* expression. End = endoderm; For = foregut in all panels. *p < 0.05, error bars represent SEM.

The following figure supplement is available for figure 2:

**Figure supplement 1**. Robust induction of NKX2.1 in foregut endoderm with HH stimulation and FGF inhibition.

induce *SHH*, *NKX2.1* and *PAX8* by exposing foregut cultures to low and high concentrations of FGF2 (50, 500 ng/ml) (*Figure 2A*). We observed a robust concentration dependent increase in *SHH* and *PAX8* mRNA expression compared to foregut or endoderm controls, and a modest increase of *NKX2.1* expression at the highest dose of FGF2 (500 ng/ml) (*Figure 2A*). We also observed that dual NOG/SB inhibition in endoderm cultures induced robust *NKX2.1* and *PAX8* expression without adding FGF2 (*Figures 1B, 2A*). Thus, we wanted to determine if *NKX2.1* expression in foregut cultures was due to endogenous FGF and/or HH signaling. To test this, we inhibited the FGF or HH pathway with small molecules SU5402 (SU, 10 µm) and Sant-2 (10 µm) respectively (*Figure 2B–C*). Treating foregut cultures with the FGF inhibitor SU caused a significant, robust reduction in *PAX8* and a modest reduction in *SHH*, while *NKX2.1* expression was unchanged compared to foregut controls (*Figure 2B*). Conversely, inhibition of HH signaling caused a significant reduction in *NKX2.1* expression, but not *PAX8* compared to untreated foregut. When FGF2 was added to the cultures, we observed a modest increase in *NKX2.1* expression, and when FGF was added along with Sant-2, *NKX2.1* expression was significantly reduced (*Figure 2C*). Together our results suggest a hierarchy where FGF is upstream of SHH and PAX8, and where SHH is upstream of NKX2.1. To test if HH signaling was able to induce NKX2.1 in foregut cultures, we added the Smoothened agonist, SAG (1 µM) to foregut cultures. The addition of SAG induced a 6.5-fold increase of *NKX2.1* expression above foregut controls (*Figure 2D*). However, SAG alone did not reduce *PAX8* expression.

Based on these results, we further hypothesized that enhancing HH signaling would result in increased *NKX2.1* expression downstream of FGF, and that simultaneous inhibition of FGF signaling would reduce PAX8 expression; therefore, we inhibited endogenous FGF signaling with SU while activating HH with SAG (*Figure 2D*). This combination caused an additional increase in *NKX2.1* expression (21-fold vs 6.5-fold with SAG only, when compared to foregut) and a concomitant decrease in *PAX8* mRNA (*Figure 2D*). Importantly, immunofluorescence was correlated with QRT-PCR data showing an increased number of NKX2.1+ cells with the addition of SAG only. SAG+SU treated cultures showed a further increase in the number of NKX2.1 expressing cells, with ~77% of all cells expressing NKX2.1 compared to ~20% in foregut controls, and almost no PAX8 expressing cells (*Figure 2—figure supplement 1*). SAG and SAG+ SU treated cells also co-expressed FOXA2 and SOX2 confirming their endodermal origin (*Figure 1—figure supplement 1*).

## HH-induced foregut spheroids give rise to human lung organoids (HLOs)

Based on the observations that stimulating HH and inhibiting FGF signaling strongly enhances NKX2.1 expression while reducing PAX8 expression (*Figure 2*), we tested multiple conditions of HH activation and FGF inhibition to induce NKX2.1^HI/PAX8^LO foregut spheroids (NOG/SB/F/Ch) (Summarized in *Figure 3—figure supplement 1*). Consistent with the important roles of FGF signaling in lung growth and branching morphogenesis (*Hebrok et al., 1998*; *Min et al., 1998*; *Weaver et al., 2000*; *Abler et al., 2009*; *Morrisey and Hogan, 2010*; *Rankin and Zorn, 2014*), we found that conditions where FGF inhibition was used led to a reduction of epithelial tissue relative to mesenchymal tissue, which could be due to a loss of epithelium or an overgrowth of mesenchyme; this suggests that endogenous FGF signaling is necessary to maintain the epithelial tissue in 3D cultures (*Figure 3—figure supplement 2*). Therefore, we also tested several conditions that stimulated HH signaling using SAG only, without FGF inhibition. We found that the most efficient method to enhance NKX2.1 expression was by adding SAG during the foregut spheroid phase (*Figure 3A*). Comparing foregut spheroids (NOG/SB/F/Ch) with those treated with SAG (NOG/SB/F/Ch/SAG), we observed a substantial decrease in *SOX2* expression compared to NOG/SB/F/Ch spheroids and a significant increase in

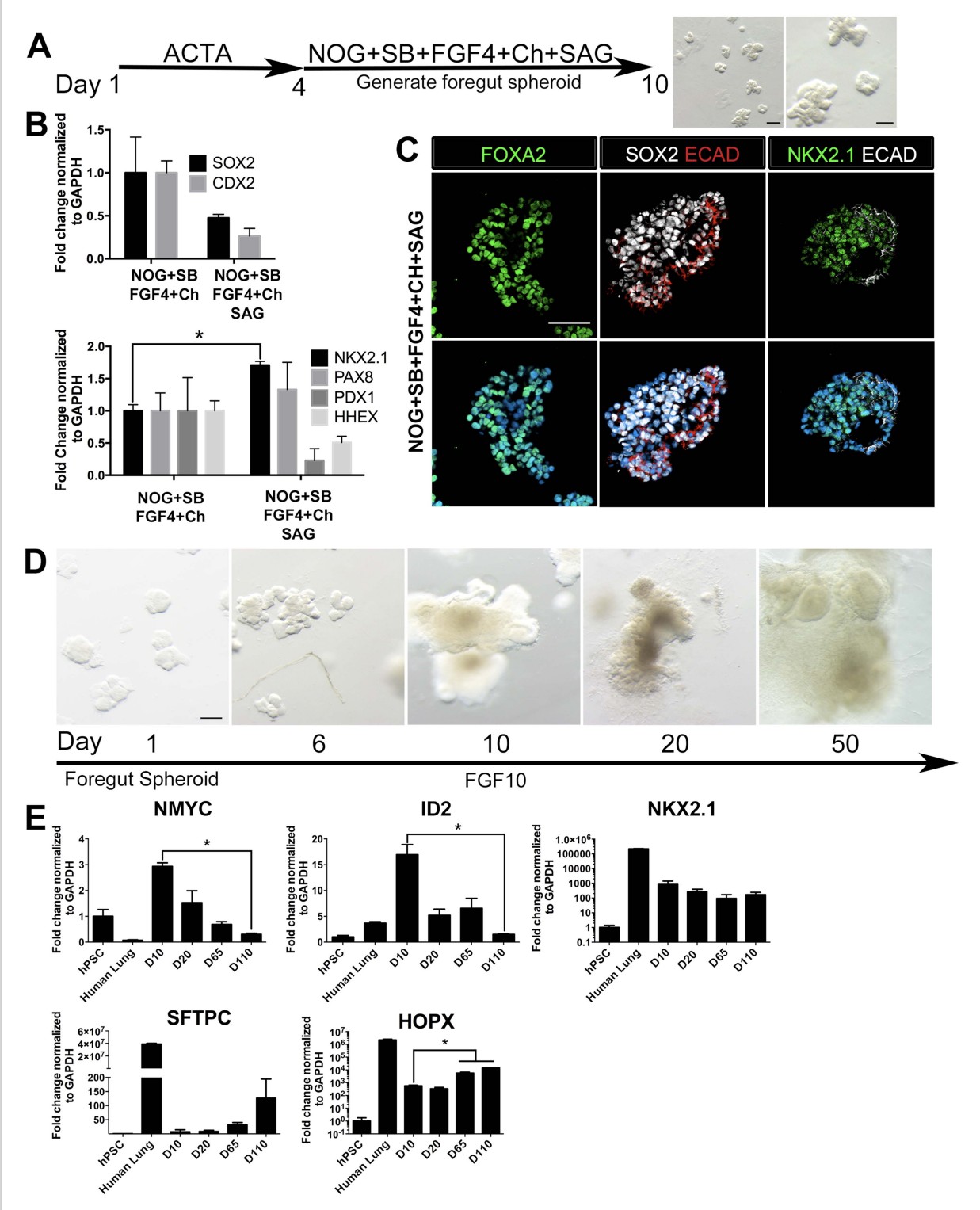

**Figure 3**. HH-induced ventral foregut spheroids give rise to lung organoids. (**A**) hESCs were differentiated into foregut spheroids by treating cells with 4 days of ACTA and then an additional 4–6 days of NOG+SB+FGF4+Ch with the addition of the HH agonist SAG. Representative whole mount images of spheroids in a matrigel droplet are shown at low (left, scale bar 200 µm) and high magnification (right, scale bar 100 µm). (**B**) The addition of SAG to the NOG+SB+FGF4+Ch spheres caused a reduction in *SOX2* and *CDX2* transcripts (top panel) and a significant increase of *NKX2.1* transcript (bottom panel) compared to NOG+SB+FGF4+Ch spheres (without SAG). Other foregut lineages (*PAX8, PDX1, HHEX*) were not significantly different when SAG was added. (**C**) The majority of the cells in NOG+SB+FGF4+Ch+SAG spheres expressed FOXA2, SOX2 and NKX2.1 protein. Scale bars represent 50 µm.
*Figure 3. continued on next page*

*Figure 3. Continued*

(**D**) Timeline showing NOG+SB+FGF4+Ch+SAG induced foregut spheroids grown and maintained in FGF10. Note that Day 1 is the day spheroids were plated in Matrigel. The scale bar represents 100 μm. (**E**) Organoids express lung markers in a manner consistent with mouse lung development. All expression is shown relative to undifferentiated pluripotent stem cells (hPSC), and adult human lung is shown as a reference. Lung progenitor markers *NMYC* and *ID2* were very low in adult lung, and were expressed at high levels in early organoid cultures, but were reduced over time (D = Days in culture), whereas *NKX2.1* expression remained relatively constant. In contrast, *SFTPC* is known to be expressed at low levels in distal lung progenitors, but increases and is highly expressed in AECII cells. Consistently, *SFTPC* is highly expressed in adult human lungs and increases over time in organoid cultures and the AECI marker *HOPX* is also highly expressed in adult human lung and increases over time in organoids. *p < 0.05. All error bars represent SEM.

The following figure supplements are available for figure 3:

**Figure supplement 1**. Overview of conditions tested to generate human lung organoids.

**Figure supplement 2**. FGF-low culture conditions cause a loss of organoid epithelium over time.

**Figure supplement 3**. Foregut spheroids express lung and foregut specific markers.

**Figure supplement 4**. Ventral foregut spheroids do not express appreciable levels of PAX8 protein.

**Figure supplement 5**. Foregut spheroids consist of both epithelial and mesenchymal cells.

**Figure supplement 6**. Lung organoids contain both proximal and distal domains.

*NKX2.1* mRNA. Additionally, nuclear NKX2.1 protein expression was found in ECAD+ epithelium which co-expressed endoderm epithelial markers FOXA2 and SOX2 (*Figure 3B,C*, *Figure 3—figure supplement 3*). Interestingly, during lung specification in mice, the gut tube initially expresses Sox2 throughout the endoderm, but Sox2 is down-regulated in the lung field during lung specification and Nkx2.1 induction (*Hebrok et al., 1998*; *Serls, 2004*; *Domyan et al., 2011*). Thus, concomitant down-regulation of *SOX2* and increased *NKX2.1* observed in SAG treated foregut spheroids is consistent with early transcriptional changes that occur during lung specification in mice.

We also observed a slight, but non-significant increase in *PAX8* transcript level in NOG/SB/F/Ch/SAG treated foregut spheroids (*Figure 3B*). Importantly, PAX8 protein expression was undetectable in NOG/SB/F/Ch/SAG treated foregut spheroids and expression remained low/undetectable throughout time in culture. (*Figure 3—figure supplement 4*). Similar to NOG/SB/F/Ch treated spheroids, the NOG/SB/F/Ch/SAG treated spheroids had a minor population of cells within the spheroids of mesodermal in origin, expressing Vimentin (VIM) (*Figure 3—figure supplement 5*).

NOG/SB/F/Ch/SAG treated foregut spheroids were embedded in Matrigel to provide a 3D growth environment. Spheroids maintained in basal media (see 'Materials and methods') supplemented with 1% FBS lost ECAD+ epithelial structures and were mainly comprised of mesenchyme within 20 days of 3D culture (*Figure 3—figure supplement 2D,E*). FGF10 is essential for branching morphogenesis and maintenance of lung progenitor cells during development as well as tissue homeostasis in the adult lung (*Bellusci et al., 1997a*; *Min et al., 1998*; *Weaver et al., 2000*; *Volckaert et al., 2013*). We observed that the addition of FGF10 (500 ng/ml) allowed spheroids to expand and be passaged for over 100 days. FGF10 promoted the maintenance of ECAD+ epithelial structures with less mesenchymal contributions compared to both basal and FGF inhibitor conditions (*Figure 3D*). NOG/SB/F/Ch/SAG cultured for 15 days in FGF10 possessed abundant ECAD+ epithelium that expressed the proximal lung marker SOX2 and distal lung marker SOX9. SOX2+ domains and SOX9+ domains were distributed throughout the entire HLO as determined by whole mount immunofluorescence and confocal Z-sections. (*Figure 3—figure supplement 6*). FGF10 treated foregut spheroids maintained *NKX2.1* expression over time; however, consistent with mouse development, distal progenitor markers, *NMYC* and *ID2* mRNA expression decreased over time while distal Alveolar Type I and II cell markers, *HOPX* and *SFTPC* increased over time (*Okubo, 2005*; *Rawlins et al., 2009*) (*Figure 3E*). These data suggest that HLOs pass through a stage resembling early fetal lung development in mice.

## HLOs possess proximal airway-like structures

HLOs cultured longer than 2 months had striking epithelial structures resembling proximal airways, expressing proximal cell type-specific markers, including basal cells (P63), ciliated cells (FOXJ1, ACTTUB) and club cells (SCGB1A1) (*Figure 4*). Proximal-like airway tissues were often surrounded by a smooth muscle actin positive (SMA+) mesenchyme compartment. Although *P63* mRNA expression is maintained throughout culture (*Figure 4A*), it is only in prolonged cultures (>2 months) where the P63+ cells are spatially arranged along the basal side of the epithelial tube-like structures, adjacent to SMA+ mesenchyme, similar to human bronchi and bronchioles (*Figure 4B*) (*Boers et al., 1998*; *Nakajima et al., 1998*; *Evans et al., 2001*; *Rock et al., 2009*). By 65 days in vitro (D65) proximal-like epithelial structures form a cyst-like structure that expresses P63, as determined by whole mount immunofluorescence staining and confocal z-stacks. Moreover, SMA expression is strongest at the periphery of the HLO (*Figure 4—figure supplement 1*). P63+ proximal airway-like cells also co-express SOX2 and NKX2.1 as determined on serial sections (*Figure 4—figure supplement 2*). Located on the luminal surface of HLO proximal airway-like structures are cells expressing the multi-ciliated cell transcription factor FOXJ1 (*Figure 4B*). Very few cells expressed the club cell marker SCGB1A1, and this protein was observed in a pixilated expression pattern (*Figure 4D*). Multi-ciliated and club cell specific mRNAs, *FOXJ1* and *SCGB1A1* respectively, were significantly increased in prolonged HLO culture (*Figure 4A*). Although the goblet cell marker *MUC5AC* mRNA expression was detected, protein expression was not detected by immunofluorescence (*Figure 4A* and data not shown).

Although the multi-ciliated cell transcription factor FOXJ1 was abundant in proximal airway-like structures, we observed that ACTTUB was localized to the apical side of these cells, but did not appear to be localized to cilia on the apical cell surface (*Figure 4C*), suggesting that this may represent a cell that has not fully differentiated. Others have demonstrated that robust differentiation of multi-ciliated cells from hPSCs require modified culture conditions to promote differentiation of functional cell types (*Firth et al., 2014*). Thus, it is possible that the HLO environment, such as Matrigel or media rich in FGF10, does not promote terminal differentiation of all cell types. In order to alter the HLO environment, we seeded NOG/SB/F/Ch/SAG foregut spheroids onto an acellular human lung matrix (*Booth et al., 2012*). Spheroids seeded on slices of acellular lung matrix predominantly gave rise to proximal airway-like structures in which stereotypical tufts of ACTTUB positive ciliated structures on the apical surface of cells were observed facing into a lumen. In serial sections, these airways had abundant FOXJ1+ cells (*Figure 4E*). Thus, HLOs have the capacity to generate more mature ciliated cells given the correct stimulus or environment.

As noted, proximal airways are often closely associated with the SMA+ mesenchyme (*Figure 4B*) whereas in the adult murine lung, proximal airways are also associated with Pdgfrα+ and Vim+ mesenchymal cells (*Boucherat et al., 2007*; *Hinz et al., 2007*; *Chen et al., 2012*). Thus, we investigated the mesenchymal population within the HLOs in more detail. Immunofluorescence revealed that D65 HLOs have both PDGFRα+/VIM+ double positive and PDGFRα−/VIM+ cell populations, which is indicative of myofibroblasts and fibroblasts respectively (*Figure 5A*). Adult murine myofibroblasts also co-express Sma and Pdgfrα whereas differentiated smooth muscle is Sma+/Pdgfra− (*Leslie et al., 1990*; *Low and White, 1998*; *Boucherat et al., 2007*; *Hinz et al., 2007*; *Chen et al., 2012*), and we observe PDGFRα+/SMA+ and PDGFRα−/SMA+ populations of cells indicating that HLOs possess myofibroblasts and smooth muscle cells (*Figure 5B*). The HLOs did not stain positive for Safranin O indicating there is no cartilage tissue, whereas iPSC derived teratomas had abundant cartilage (*Figure 5C*). Taken together, the HLO mesenchymal population is diverse with myofibroblasts, fibroblasts, and smooth muscle cells.

## HLOs possess immature alveolar airway-like structures

The distal lung epithelium in mouse and human make up the gas-exchanging alveoli, consisting of type I and type II alveolar epithelial cells (AECI, AECII). During development, the distal lung epithelium expresses progenitor markers including SOX9, ID2, and NMYC (*Okubo, 2005*; *Rawlins et al., 2009*; *Chang et al., 2013*; *Rockich et al., 2013*). All distal markers are present in the HLOs; however, *ID2* and *NMYC* are expressed at high levels in early cultures, but are down regulated in prolonged culture (*Figure 3F*) while *SOX9* expression remains consistent across time in culture (*Figure 6A*).

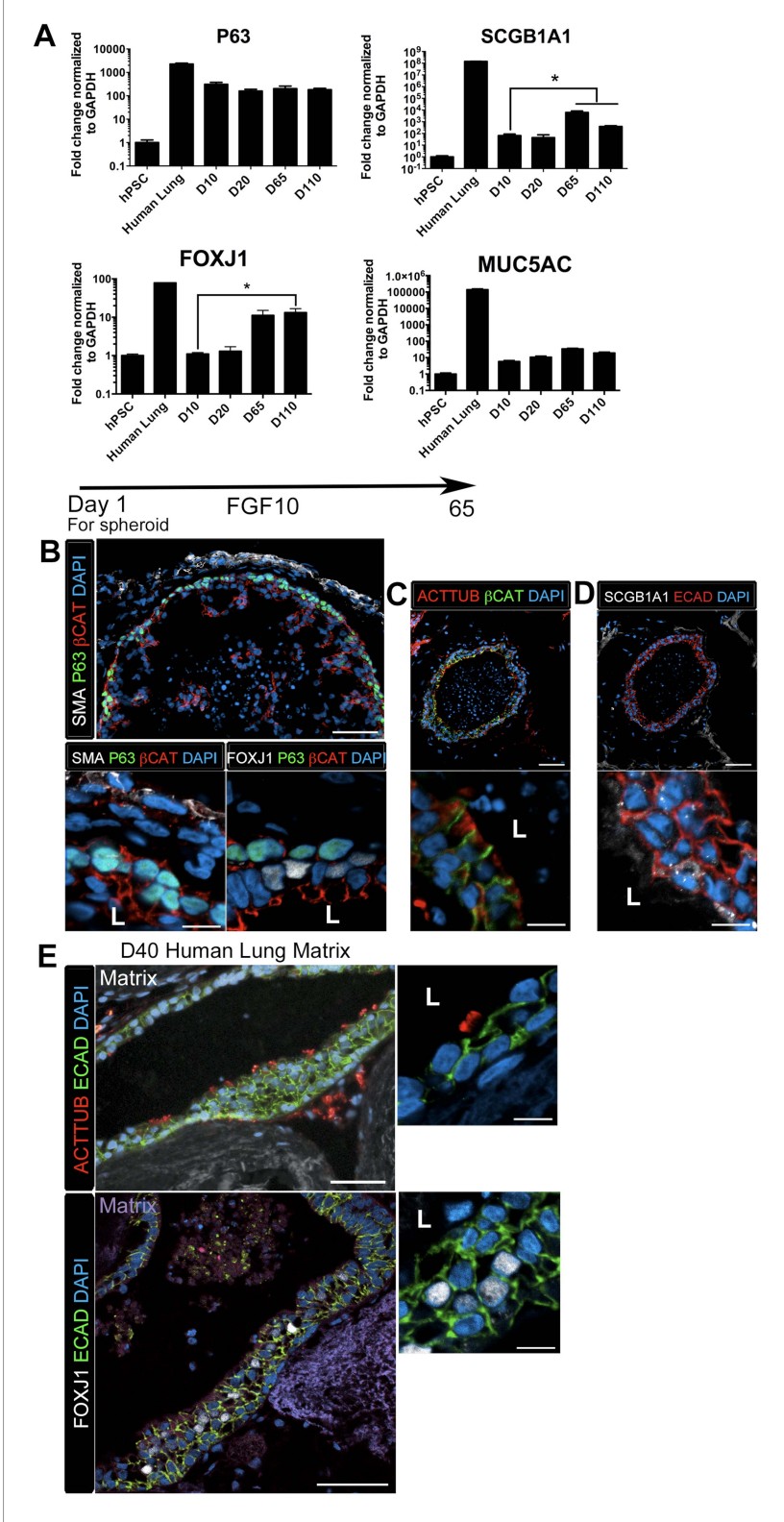

**Figure 4**. Lung organoids form proximal airway-like structures. (**A**) Genes expressed in the proximal airway were examined in organoids across time. The proximal airway cell marker *SOX2* decreased over time in HLOs cultures compared to D10 HLOs. Compared to undifferentiated hPSCs, organoids expressed high levels of the basal cell marker *P63* at all time points, while expression of the club cell marker *SCGB1A1* and ciliated cell marker *FOXJ1*

*Figure 4. continued on next page*

*Figure 4. Continued*

increased significantly in prolonged cultures (compared to D10 HLOs). There was an increasing but non-significant trend in goblet cell *MUC5AC* expression over time in culture. (**B**) D65 HLOs had structures resembling the proximal airway, in which the epithelium (β-catenin, red) possesses P63+ basal cells (green), and is surrounded by SMA+ (white, upper and lower left panel) mesenchymal tissue. Adjacent to the P63 positive basal cell layer (green, lower, right panel) were FOXJ1 positive cells (white). Scale bars represent 50 µM (top) and 10 µM (bottom). (**C**) Proximal airway-like epithelium (β-catenin, green) co-stained for ACTTUB on the apical side of the cell (red). Scale bars represent 50 µM (top) and 10 µM (bottom). (**D**). Proximal airway-like epithelium (E-cadherin, red) also co-stained with Club cell marker CC10 (white, right panel). Scale bars represent 50 µM (top) and 10 µM (bottom). (**E**) Acellular human lung matrix was seeded with spheroids and cultured for 40 days (D40). Matrices had abundant proximal airway-like structures that had multi-ciliated cells on the apical surfaced labeled by ACTTUB (red, top panel) in low (scale bar 50 µM) and high magnification (scale bar 10 µM). Serial sections showed that cells were also FOXJ1 positive (white, lower panel) with the epithelium outlined in ECAD (green) in low (scale bar 50 µM) and high magnification (scale bar 10 µM). (**B–D**) 'L' in high magnification images indicates the lumen. *p < 0.05. All error bars represent SEM.

The following figure supplements are available for figure 4:

**Figure supplement 1**. Lung organoids have P63+ epithelium throughout the organoid.

**Figure supplement 2**. P63+ cells have an NKX2.1+ lung identity.

Recently, there have been major advances in mice toward defining a bipotent alveolar progenitor population during the late fetal/early neonatal period (*Desai et al., 2014*; *Treutlein et al., 2014*), and this work has highlighted the fact that many markers previously considered terminal differentiation markers are co-expressed in the bipotent progenitors. Specifically, the AECII marker SftpC and AECI marker Hopx can be co-expressed in a bipotent progenitor before becoming committed to one lineage or the other. Moreover, we have shown that Sox9 marks an early progenitor population in the developing mouse lung and Sox9 also marks the bipotent progenitor in late fetal life (*Rockich et al., 2013*; *Treutlein et al., 2014*). In HLOs grown in prolonged culture (>2 months), we observed that AECII (SFTPC, SFTPB) and AECI (PDPN, HOPX) cell-type markers were present (*Figure 6A–B*). However, we also observed that *SFTPC* levels were very low (*Figure 3F*), and that SFTPB+ cells were rare (*Figure 6B*). This suggested that the distal airway cells present in HLOs might be a progenitor-like population. To test this possibility, we co-stained SFTPC (AECII) or HOPX (AECI) with SOX9 and found abundant SFTPC/SOX9 and HOPX/SOX9 double positive cells (*Figure 6B*). Co-staining in serial sections suggests that SFTPC/SOX9 double positive cells are also NKX2.1+ (*Figure 6—figure supplement 1*). In contrast these co-expressing cells were not found in the adult human lung (*Figure 6C*). Although rare, the few SFTPB+ observed in HLOs resemble AECII cells seen in the adult human lung, and PDPN+ cells resembled the elongated AECI cells in the human lung (*Figure 6B–C*). In order to improve confidence that cells expressing AECII markers are AECII cells, we used transmission electron microscopy (TEM) to determine if HLOs possessed cells containing lamellar bodies, which are necessary for surfactant protein trafficking and secretion (*Schmitz and Müller, 1991*; *Stahlman et al., 2000*; *Weaver et al., 2002*). Using TEM, we observed lamellar bodies both in cells within HLOs, and in open spaces between cells, indicating that lamellar bodies are being secreted (*Figure 6D*). Taken together, our data suggests that HLOs predominantly possess undifferentiated alveolar progenitor cells with rare differentiated AECI and AECII cells interspersed throughout the distal-like tissue.

## Quantitative assessment of HLO composition

We have shown that HLOs have both proximal-like and distal-like epithelium in addition to surrounding mesenchymal tissue. In order to better gauge the composition of HLOs, we performed a detailed quantitative analysis of cell types and structures. We sectioned 48 individual HLOs, and examined them for P63+ proximal airway-like structures (as shown in *Figure 4B–D*), and distal-airway like structures (as shown in *Figure 6—figure supplement 1*). We found that 39/48 (81%) of the HLOs have proximal airway epithelial structures while 48/48 (100%) of HLOs have distal airway-like structures (*Figure 7A*). We then calculated the average cross-sectional area comprised of P63+

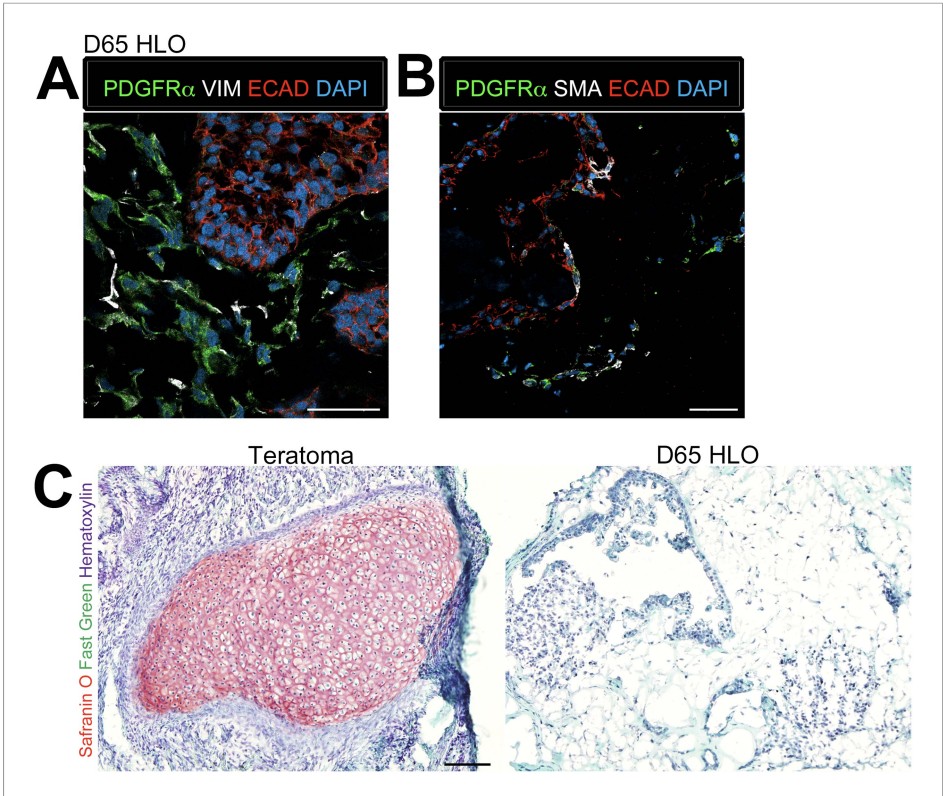

**Figure 5**. Lung organoids possess multiple types of mesenchymal cells. (**A**) D65 HLOs have PDGFRα+ (green) VIM+ (white) double-positive myofibroblasts and PDGFRα–/VIM+ fibroblasts. Scale bar represents 50 μm. (**B**) D65 HLOs also possesses PDGFRα+ (green) SMA+ (white) double-positive myofibroblasts and PDGFRα–/SMA+ smooth muscle and myofibrblasts. Scale bar represents 50 μm. (**C**) D65 HLOs do not contain any cartilage whereas positive control iPSC derived teratoma had clear Safranin O staining specific to cartilage. Fast green marks the cytoplasm and hematoxylin the nuclei of both tissues. Scale bar represents 100 μm.

proximal airway-like and P63–/SFTPC+ distal airway-like tissue and found that proximal structures comprised 14.5% (±0.6%) of the entire area of the HLO, whereas 85.5% (±0.6%) were distal in nature (including epithelium and mesenchyme) (*Figure 7B*). To determine the percentage of certain cell types within an HLO, we sectioned and stained 15 individual HLOs (n = 15) and counted cells positive for specific markers, and the total number of Dapi+ nuclei within a section (*Figure 7C–G*). On average, 57% of all cells in the HLOs were NKX2.1+ (*Figure 7C*), 39% of all cells were P63+, 3% were FOXJ1+, 5% were SFTPC+, and 4% of all cells were HOPX+ (*Figure 7D–G*).

## HLOs are globally similar to human fetal lung

Accumulating evidence suggests that HLOs are immature. For example, distal progenitor markers are initially robustly expressed whereas SFTPC expression is very low across time in HLOs (*Figure 3E*), FOXJ1+ cells do not appear to form mature multi-ciliated structures until placed onto a decelluarized lung matrix (*Figure 4B,E*) and rare SCGB1A1+ cells do not resemble mature club cells (*Figure 4D*). Moreover, the majority of the distal-like epithelium expresses bipotent progenitor markers (*Figure 5*). In order to directly address the maturity of HLOs, we conducted RNA-sequencing (RNAseq) on HLOs (n = 6; 3 D65 HLOs, 3 D110 HLOs), on undifferentiated hESCs and on definitive endoderm. We also took advantage of publicly available RNAseq datasets for human fetal lung representing a range of gestational stages, and for adult human lung (*Supplementary file 1*). In order to determine global similarity among these tissues relative to HLOs, we conducted principal component (PC) analysis (*Figure 8A,B*) (*Ringnér, 2008*), hierarchical clustering (*Figure 8C*) (*Eisen et al., 1998*) and Spearman's rank-order correlation (*Jiang et al., 2004*) matrix analysis (*Figure 8D*) of the complete tabulated FPKM matrix generated from RNA sequences datasets and representing the total gene expression

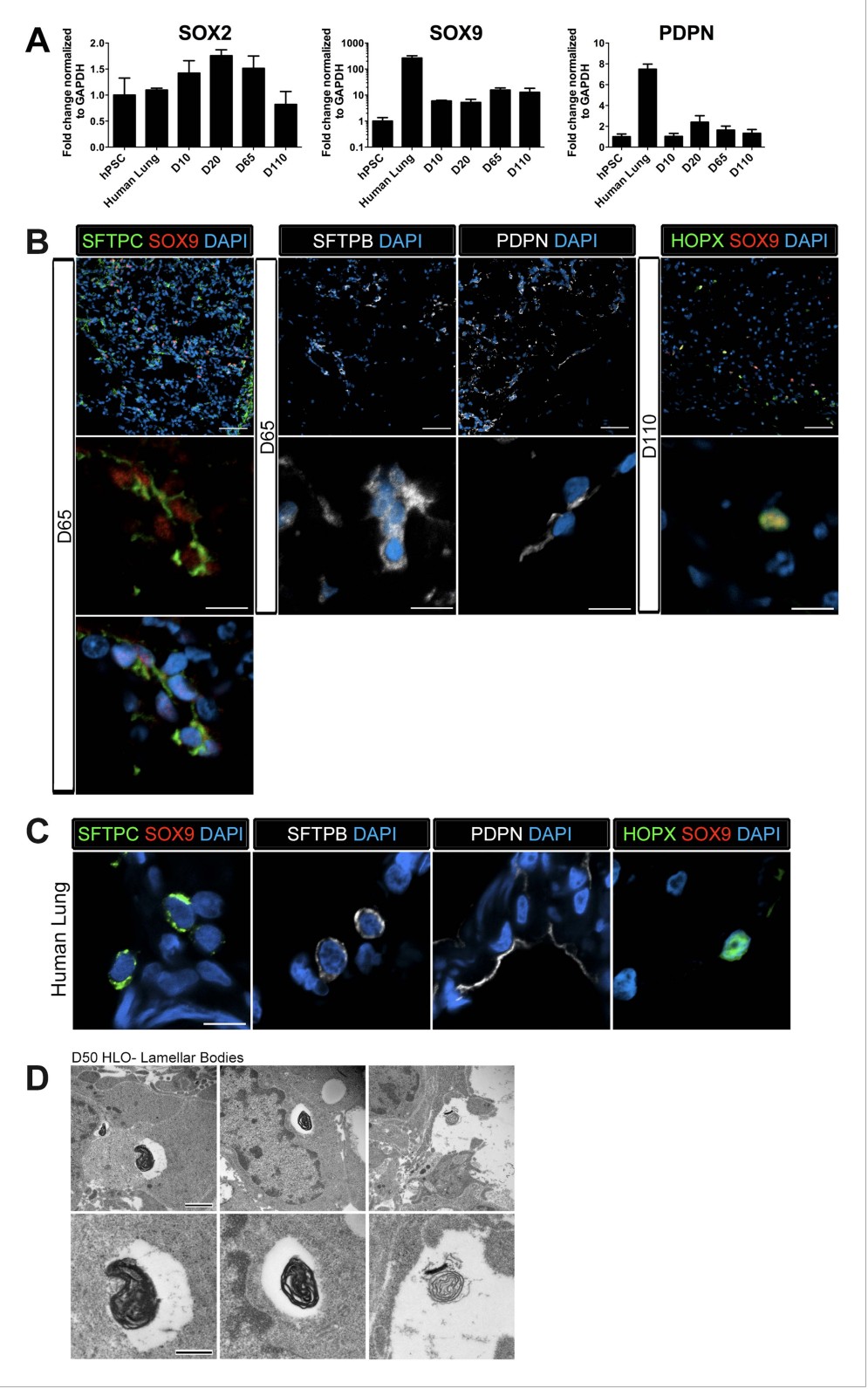

**Figure 6**. Lung organoids possess abundant distal bipotent progenitor cells. (**A**) The expression of the distal progenitor marker *SOX9* remained unchanged over time and expression of the AECI marker *PDPN* was low in HLO cultures. (**B**) The majority of SFTPC+ cells (green, left panel) co-expressed SOX9 (red). Similarly, many cells expressing the AECI early marker HOPX+ (green, right panel) co-expressed SOX9 (red). Few, scattered cells

*Figure 6. continued on next page*

*Figure 6. Continued*

expressed the late AECII marker SFTPB (white, second panel) or the AECI marker, PDPN (third panel, white). Few PDPN+ cells also showed elongated, squamous morphology seen in the adult lung. (**C**) Human lung AECII cells labeled with SFTPC (green, left panel) did not co-express SOX9. SFTPB+ cells (white, second panel) in the adult human lung have similar morphology to SFTPB+ cells in HLOs. Human lung AECI cells expressed PDPN (white, third panel), and show characteristic AECI cell shape. Human AECI cells express HOPX (green, right panel), but did not co-express SOX9. (**B–C**) Scale bar in lower magnification images in B (upper panel) represent 50 µM and the scale bars in higher magnification images in **B**, **C** (lower panel) represent 10 µM. (**D**) D50 HLOs contain lamellar bodies which are organelles specific to AECII cells. Scale bars represent 500 nm.

The following figure supplement is available for figure 6:

**Figure supplement 1**. SFTPC+ cells express lung specific markers.

complement in each sample. Consistent across all three types of informatics analysis, transcriptional activity in the HLOs shares the greatest degree of similarity to human fetal lung. These data strongly suggest that global transcription of HLOs is highly similar to human fetal lung, and support the idea that HLOs are in a less differentiated, fetal state when grown in the conditions described here.

## Discussion

To date, a number of groups have defined methods to generate lung specific cell types utilizing 2D culture systems (*Green et al., 2011*; *Longmire et al., 2012*; *Mou et al., 2012*; *Wong et al., 2012*; *Ghaedi et al., 2013*; *Huang et al., 2013*). Although lung lineage cells have been generated with varying efficiency (∼30–80% NKX2.1+ cells [*Wong et al., 2012*; *Huang et al., 2013*; *Firth et al., 2014*]) and can generate both both proximal (∼5–36% of cells [*Wong et al., 2012*; *Huang et al., 2013*; *Firth et al., 2014*]) and distal cell types (up to ∼50% of cells [*Huang et al., 2013*]), proper spatial organization of the cell types and specific tissue morphology have not been reported in 2D systems. Here, we show that HLOs possess both mesenchymal and lung epithelial (∼60% NKX2.1+) cells with proximal airway-like structures that possess P63+ (∼40%) and FOXJ1+ cells (∼3%) along with distal airway-like structures that possess SFTPC+ (∼5%) and HOPX+ (∼4%) cells.

It is currently unclear if 2D culture systems described have the capability to give rise to mesodermal lineages. Thus, HLOs allow one to address questions regarding spatial tissue organization and epithelial-mesenchymal interactions. Since HLOs form organized structures that resemble bronchi and bronchioles with adjacent mesenchyme, these complex, organized tissues may allow exploration, for example, of airway remodeling after injury. Moreover, the spatial arrangement of specific cell types will be critical to study proximal airway dynamics during homeostasis and injury. For example, the location of P63+ cells adjacent to FOXJ1+ cells in the HLOs will be necessary to study basal cell differentiation into different proximal airway cell types during homeostasis or after injury. In addition to tissue morphology and structure during prolonged culture, the HLOs consist of both epithelium and mesenchyme in early cultures that are maintained over time. Since lung development requires extensive cross talk between the epithelium and mesenchyme in order to regulate developmental processes, proliferation and differentiation, HLOs may be an ideal in vitro system to study these complex tissue–tissue interactions.

Recently, there has been a push to define progenitor populations during lung development and adult homeostasis in order to better understand differentiation and the transition between branching and alveolarization. Two groups have defined a bipotent progenitor population in the embryonic/neonatal lung that gives rise to both AECI and AECII cells (*Desai et al., 2014*; *Treutlein et al., 2014*). These bipotent cells express the distal progenitor marker Sox9 along with differentiation markers of AECI and AECII cells, including SftpC, HopX, and Pdpn. We demonstrate that HLOs expressed both AECI and II markers; however, the majority of these cells also expressed SOX9 suggesting that the majority of the distal epithelium is comprised of bipotent progenitors. Thus, HLOs will allow us to gain insight into this bipotent population, explore how bipotent progenitors are regulated, and define the mechanisms of how fate-decisions are made as terminal differentiation occurs.

The evidence supporting that HLOs are fetal in nature could reflect the fact that a block to full maturation exists in vitro, as is the case with other endoderm lineage organoids (intestinal and

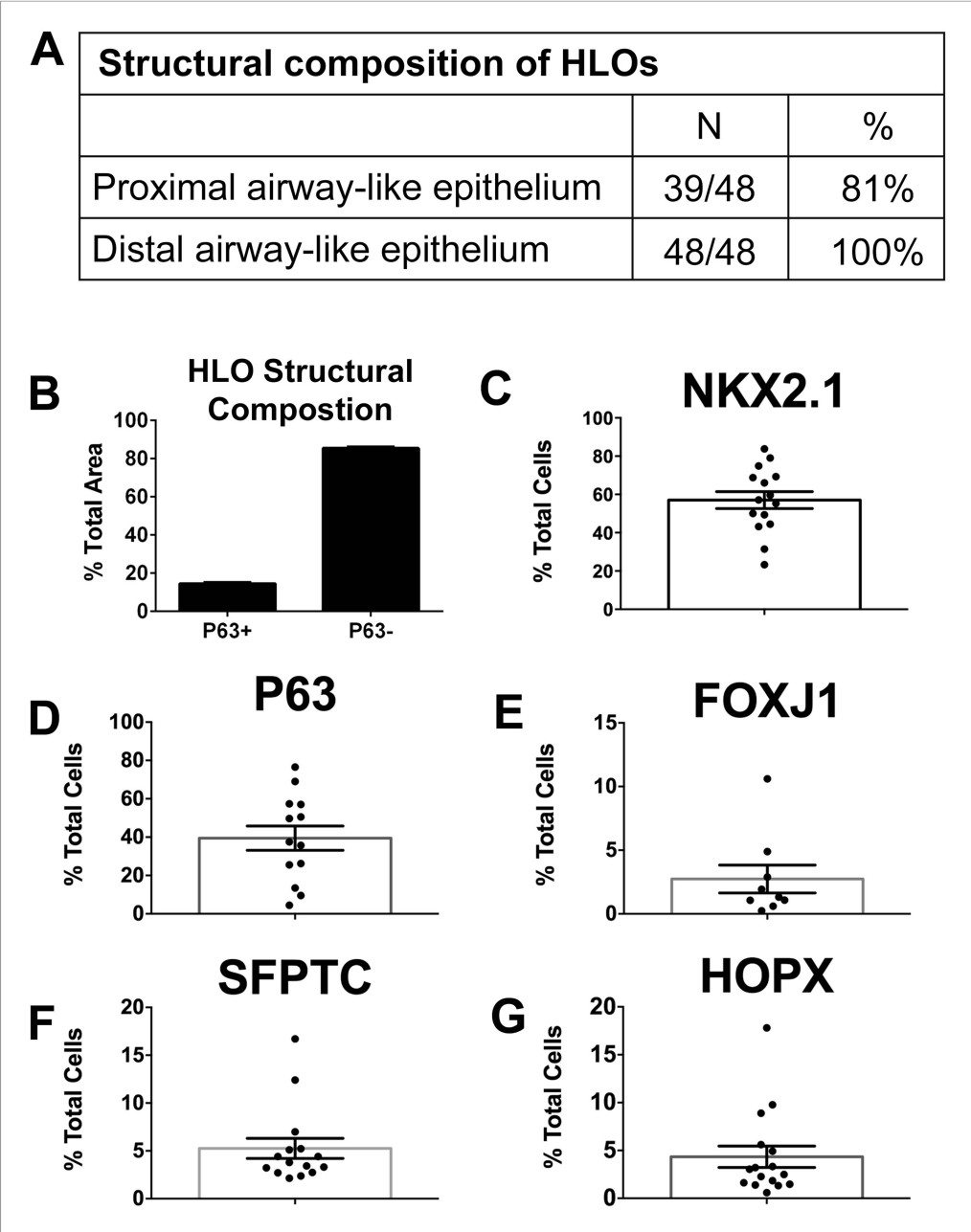

**Figure 7**. Quantitative assessment of the composition of lung organoids. (**A**) HLOs were assessed for proximal airway-like structures (P63+) and distal airway-like structures (P63–/SFTPC+). 81% of HLOs have proximal airway-like epithelium while 100% have distal airway-like epithelium (n = 48 individual HLOs). (**B**) The average cross-sectional area within an HLO that is comprised of P63+ proximal airway-like and P63–/SFTPC+ distal airway-like epithelium was calculated. Proximal structures comprised 14.5% (±0.6%) of the entire area of the HLO (P63+), whereas 85.5% (± 0.6%) of HLO was distal-like epithelium and mesenchyme (P63–). (**C–G**) The percent of specific cell markers present in an organoid was determined by dividing by the total number of DAPI+ nuclei within the same section (n = 15 individual HLOs). Each point represents the data from an individual HLO while the open bar represents the average percent of cells. (**C**) On average, 57% of all cells in the HLOs were NKX2.1+, (**D**) 39% of all cells were P63+, (**E**) 3% were FOXJ1+, (**F**) 5% were SFTPC+, (**G**) 4% of all cells were HOPX+. (**B–G**) Error bars represent SEM.

gastric), which appear to be immature. That is, while they possess committed lineage-specific cell types, the cells may not exhibit fully matured adult-like function (***McCracken et al., 2014***; ***Watson et al., 2014***). This is also the case for pancreatic β-like cells and hepatocyte-like cells generated in vitro

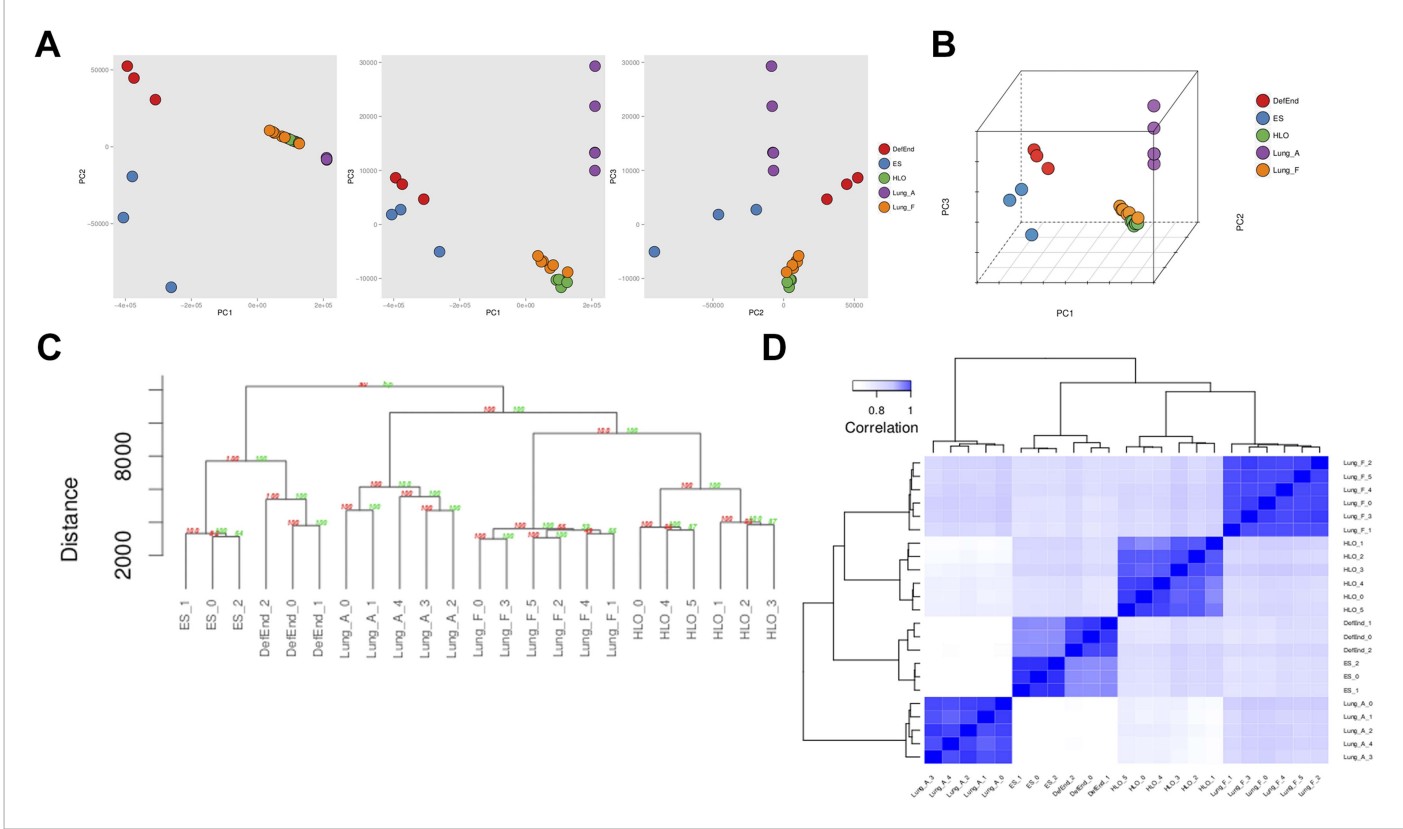

**Figure 8**. RNA sequencing analysis associates HLOs with fetal lung tissue. 6 HLOs (n = 3 D65 HLOs and n = 3 D110 HLOs) were compared to the undifferentiated H9 stem cells (SC) and definitive endoderm (Def End) and publicly available datasets of adult and fetal human lungs (see *Supplementary file 1*). (**A–B**) Plot of the first three principle components generated in the principle component (PC) analysis as pairwise 2-dimensional plots (**A**) or as an aggregate 3-dimensional projection (**B**), (**C**) hierarchical clustering, and (**D**) Spearman's correlation all demonstrate that HLOs are most closely related to the fetal lung.

(*Si-Tayeb et al., 2009*; *Hrvatin et al., 2014*). Alternatively, the progenitor state may reflect the high levels of FGF10 in the culture media, since FGF10 is known to maintain progenitor cells in the lung (*Ramasamy et al., 2007*; *Nyeng et al., 2008*). Given that HLOs are similar to human fetal lung, this tissue is an ideal model to study lung maturation of both the proximal and distal epithelium along with epithelial-mesenchymal interactions in a developmental context.

While the multi-lineage, multi-cellular composition of HLOs is a major advantage, one of the caveats to this system is that HLOs do not appear to undergo *bona fide* branching morphogenesis or possess transitional zones found in the adult lung, such as the bronchioalveolar ductal junction (BADJ). The HLOs possess proximal SOX2+ domain and distal SOX9+ domains observed during branching morphogenesis, but this regionalization occurs without setting up the stereotyped branching pattern. This may be due to the fact that the organoids are surrounded by media supplemented with FGF10 compared to the in vivo situation where FGF10 is expressed in a dynamic, spatially restricted manner in the distal mesenchyme (*Bellusci et al., 1997b*; *Nyeng et al., 2008*; *Abler et al., 2009*). However, it has recently been demonstrated that localized expression FGF10 is not required for branching (*Volckaert et al., 2013*), so this may not explain the lack of branching. Alternatively, similar to other endoderm-derived organoid models, HLOs lack several components of the native organ, including immune cells, vasculature, and innervation. Thus, it is possible that cellular inputs important for branching morphogenesis are missing from HLOs. Indeed, recent reports have shown that innervation is required for proper branching (*Bower et al., 2014*), and while vasculature may not be important for lung branching (*Havrilak and Shannon, 2015*), others have shown the vascular endothelium is required to induce a branching-like program of isolated airway epithelium in 3D cultures (*Franzdóttir et al., 2010*). Lastly, the microenvironment is essential for branching morphogenesis to occur including dynamic changes in the extracellular matrix around branching lung bud tips where the ECM

is constantly changing and interacting with the cytoskeleton of the branching epithelium in order to facilitate cell movement and branching bifurcations (*Moore et al., 2005*; *Kim and Nelson, 2012*; *Wan et al., 2013*). It is possible that in the future, co-culture with additional cellular inputs may prove to enhance HLO branching.

Taken together, we describe here a novel system to generate human lung organoids from human pluripotent stem cells. HLOs possess both mesenchymal and epithelial lineages, as well as organized proximal airway structures with multiple cell types and surrounded by mesenchyme. HLOs also possess distal epithelial cells that are reminiscent of a bipotent alveolar progenitor cell recently described in mice which is likely a reflection of the similarities of HLOs to the human fetal lung. We believe that HLOs will be an excellent new human system to model lung differentiation, homeostasis and disease in vitro.

## Materials and methods

### Maintenance of hESCs

Human ES cell lines H1 (NIH registry #0043) and H9 (NIH registry #0062) were obtained from WiCell Research Institute. Human ES line UM77-2 (NIH registry #0278) was obtained from the University of Michigan. iPSC lines 3-5 and 20-1 were generated at Cincinnati Children's Hospital and have been previously described (*Spence et al., 2011*). Stem cells were maintained on Matrigel (BD Biosciences, San Jose, CA) in mTeSR1 medium (STEMCELL Technologies, Vancouver, Canada). HESCs were passaged as previously described (*Spence et al., 2011*).

### Differentiation of PSCs into definitive endoderm

Differentiation into definitive endoderm was carried out as previously described (*D'Amour et al., 2005*; *Spence et al., 2011*). Briefly, a 4-day Activin A (R&D systems, Minneapolis, MN) differentiation protocol was used. Cells were treated with Activin A (100 ng ml$^{-1}$) for 3 consecutive days in RPMI 1640 media (Life Technologies, Grand Island, NY) with increasing concentrations of 0%, 0.2% and 2% HyClone defined fetal bovine serum (dFBS, Thermo Scientific, West Palm Beach, FL).

### Differentiation of definitive endoderm into anterior foregut

After differentiation into definitive endoderm, foregut endoderm was differentiated, essentially as described (*Green et al., 2011*). Briefly, cells were incubated in foregut media: Advanced DMEM/F12 plus N-2 and B27 supplement, 10 mM Hepes, 1× L-Glutamine (200 mM), 1× Penicillin-streptomycin (5000 U/ml, all from Life Technologies) with 200 ng/ml Noggin (NOG, R&D Systems) and 10 µM SB431542 (SB, Stemgent, Cambridge, MA) for 4 days. For long term maintenance, cultures were maintain in 'basal' foregut media without NOG and SB, or in the presence of growth factors including 50, 500 ng/ml FGF2 (R&D systems), 10 µM Sant-2 (Stemgent), 10 µM SU5402 (SU, Stemgent), 100 ng/ml SHH (R&D systems), and SAG (Enzo Life Sciences, Farmingdale, NY) for 8 days.

### Directed differentiation into anterior foregut spheroids and lung organoids

After differentiation into definitive endoderm, cells were incubated in foregut media with NOG, SB, 500 ng/ml FGF4 (R&D Systems), and 2 µM CHIR99021 (Chiron, Stemgent) for 4–6 days. After 4 days with treatment of growth factors, three-dimensional floating spheroids were present in the culture. Three-dimensional spheroids were transferred into Matrigel to support 3D growth as previously described (*McCracken et al., 2011*). Briefly, spheroids were embedded in a droplet of Matrigel (BD Bioscience #356237) in one well of a 24 well plate, and incubated at room temperature for 10 min. After the Matrigel solidified, foregut media with 1% Fetal bovine serum (FBS, CAT#: 16000–044, Life Technologies) or other growth factors and small molecules were overlaid and replaced every 4 days. Organoids were transferred into new Matrigel droplets every 10–15 days.

### Immunohistochemistry

Immunostaining was carried out as previously described (*Spence et al., 2009*; *Rockich et al., 2013*). Antibody information and dilutions can be found in *Supplementary file 2*. All images were taken on a Nikon A1 confocal microscope or an Olympus IX71 epifluorescent microscope.

## RNA extraction and qRT-PCR

RNA was extracted from monolayers, spheroids, and organoids using a MagMAX-96 Total RNA Isolation Kit (Life Technologies) and MAG Max Express (Applied Biosystems, Grand Island, NY). RNA quantity and quality were determined spectrophotometrically, using a Nano Drop 2000 (Thermoscientific). Reverse transcription was conducted using the SuperScript VILO kit (Invitrogen, Grand Island, NY), according to manufacturer's protocol. Finally, qRT-PCR was carried out using Quantitect Sybr Green MasterMix (Qiagen) on a Step One Plus Real-Time PCR system (Life Technologies). For a list of primer sequences see *Supplementary file 3*.

## Seeding lung spheroids on decellularized human lung matrices

Human lungs deemed to be unsuitable for lung transplantation were obtained from beating-heart (or warm autopsy) donors through Gift of Life Michigan and lungs were decellularized as previously described (*Booth et al., 2012*). Slices were prepared using a sterile tissue punch (Fisher) and sterilized with 0.18% peracetic acid and 4.8% EtOH. Matrix slices were placed in a 96 well plate and approximately 50 NOG+SB +F+Ch+SAG spheres were pipetted directly onto the matrices. Samples were centrifuged for 2 min at 2000 rpm and then incubated at 37°C for 30 min without media. Foregut media supplemented with 1% FBS and 500 ng/ml FGF10 was then added to the matrices. Media was changed daily.

## Transmission electron microscopy

D50 HLOs were processed as previously described (*Prasov et al., 2012*; *Rockich et al., 2013*). 70 nm sections were sections were imaged using a Philips CM-100 electron microscope.

## Area and cell quantification

HLOs with P63+ cells were counted as having proximal airway-like epithelium and HLOs with SFTPC+ cells were counted as having distal airway-like epithelium. The area of proximal epithelium was determined by P63+ECAD+ staining. Area was measured using ImageJ software. Cell quantification of NKX2.1, P63, and DAPI was counted by Metamorph cell counting software. FOXJ1, SFTPC, and HOPX were counted in ImageJ using the cell counter plugin.

## Statistical analysis and experimental replicates

All immunofluorescence and qRT-PCR experiments were carried out at least two times with three (n = 3) independent biological samples per experiment. The only exceptions to this were experiments that included human adult lung samples in the analysis. For these experiments, n = 1 biological human lung sample was used in statistical replicates (triplicates) whereas all other samples used biological replicates (n = 3). For quantification in *Figure 7*, a total of 48 different HLOs (n = 48) were counted for HLO composition. For the proximal epithelial area, 29 different HLOs were counted (n = 29). For cell quantification, 15 different HLOs were counted (n = 15). Statistical differences between groups were assessed with Prism software, using multiple t tests. All error bars represent SEM. Results were considered statistically significant at p < 0.05.

## RNA sequencing and analysis

Sequencing of HLOs (n = 3 D65, n = 3 D110) was performed by the University of Michigan DNA Sequencing Core, using the Illumina Hi-Seq platform. Sequencing of H9 Stem Cells (SC) and Definitive Endoderm (DE) was performed by the University of California, San Francisco DNA Sequencing Core using the Illumina Hi-Seq platform. All sequences were deposited in the EMBL-EBI ArrayExpress database using Annotare 2.0 and are catalogued under the accession number E-MTAB-3339 for the HLOs and E-MTAB-3158 for SC and DE. The University of Michigan Bioinformatics Core obtained the reads files and concatenated those into a single '.fastq' file for each sample. The Bioinformatics Core also downloaded reads files from EBI-AE database (Adult lung Samples) and NCBI-GEO (SRA) database (Fetal lung samples) (*Supplementary file 1*). The quality of the raw reads data for each sample was evaluated using FastQC (version 0.10.1) to identify features of the data that may indicate quality problems (e.g., low quality scores, over-represented sequences, inappropriate GC content, etc). Initial QC report indicated over-representation of Illumina adapter sequences in samples from EBI-AE data set and NCBI-GEO data set. Adapter sequences were trimmed from the reads using Cutadapt (version 0.9.5) (*Chen et al., 2014a*). Briefly, reads were aligned to the reference transcriptome

(UCSC hg19) using TopHat (version 2.0.9) and Bowtie (version 2.1.0.0) (*Langmead et al., 2009*). Cufflinks/CuffNorm (version 2.2.1) was used for expression quantitation and differential expression analysis (*Trapnell et al., 2012*), using UCSC hg19.fa as the reference genome sequence and UCSC hg19.gtf as the reference transcriptome annotation. For this analysis, we used parameter settings: '–multi-read-correct' to adjust expression calculations for reads that map in more than one locus, as well as '–compatible-hits-norm' and '–upper-quartile –norm' for normalization of expression values. Normalized FPKM tables were generated using the CuffNorm function found in Cufflinks. Transcriptional quantitation analysis in Cufflinks was conducted using the 64-bit Debian Linux stable version 7.8 ('Wheezy') platform. The complete FPKM matrix, containing frequency counts for all 24,010 genes contained in the reference genome for all 23 RNAseq samples, was evaluated using unscaled principle component analysis (PCA) to visualize and quantify multi-dimensional variation between samples (*Ringnér, 2008*). Of the 24,010 genes annotated in the reference genome, 2815 (11.7%) were not detected in the RNAseq analysis of any of the 23 samples. Principle components were calculated using the function 'prcomp' found in the R (version 3.1.2) statistical programming language (http://www.R-project.org/) and plotted using the R package 'ggplot2' (*Wickham, 2009*). Hierarchical cluster analysis based on the Canberra distance (*Eisen et al., 1998*) between FPKM vectors was used to classify discrete RNAseq samples according to the degree of total transcriptional dissimilarity indicated by the normalized FPKM values. Bootstrap analysis was used to assess the uncertainty in the assigned hierarchical clustering relationships. 10,000 bootstraping iterations were generated by repeatedly randomly sampling the FPKM dataset. The bootstrap probability (BP) of a cluster is defined as the frequency of a given relationship among the bootstrap replicates. Multiscale bootstrap resampling was used to calculate an approximately unbiased (AU) p-value for a given relationship, with AU > 95 indicating a high degree of statistical significance. Analyses were conducted using R package 'pvclust' (*Suzuki and Shimodaira, 2006*). Spearman correlation was applied as an additional assessment of the cumulative degree of correlation among RNAseq datasets. In addition, we computed Spearman rank correlation coefficients ($\rho$) in a pairwise manner among all 23 RNAseq samples using the complete normalized FPKM data. The Spearman coefficients were plotted as a heatmap using the function 'heatmap.2' in the R package 'gplots' (http://CRAN.R-project.org/package=gplots). Complete data analysis scripts are available at https://github.com/hilldr/HLO_eLife2015.

## Acknowledgements

JRS is supported by the NHLBI (R21HL115372, R01HL119215), and by career development grants from the NIDDK (K01DK091415) and the March of Dimes (Basil O'Connor starter scholar award).

## Additional information

### Funding

| Funder | Grant reference | Author |
| --- | --- | --- |
| National Heart, Lung, and Blood Institute (NHBLI) | R21HL115372 | Jason R Spence |
| March of Dimes Foundation | Basil O'Connor starter scholar award | Jason R Spence |
| National Institute of Diabetes and Digestive and Kidney Diseases (NIDDK) | K01DK091415 | Jason R Spence |
| National Heart, Lung, and Blood Institute (NHBLI) | R01HL119215 | Jason R Spence |

The funders had no role in study design, data collection and interpretation, or the decision to submit the work for publication.

### Author contributions

BRD, JRS, Conception and design, Acquisition of data, Analysis and interpretation of data, Drafting or revising the article; DRH, MAHF, YHT, MSN, Acquisition of data, Analysis and interpretation of data, Drafting or revising the article; RD, Acquisition of data, Contributed unpublished essential data or reagents; JMW, Conception and design, Edited and approved the final manuscript; CNM,

Conducted teratoma assays, Provided teratoma tissue, Editing and approving manuscript; RN, ODK, Conducted experiments to generate RNAseq data (ES cells, Definitive Endoderm), Conducted RNA sequencing and provided the data files, Edited and approved the final manuscript; ESW, Provided critical intellectual and material contributions to the study; GHD, Conception and design

## Additional files

### Supplementary files

• Supplementary file 1. Publicly available RNAseq datasets for human fetal lung representing a range of gestational stages and for adult human lung.

• Supplementary file 2. Antibody information and dilutions.

• Supplementary file 3. Primer sequences.

### Major datasets

The following datasets were generated:

| Author(s) | Year | Dataset title | Dataset ID and/or URL | Database, license, and accessibility information |
|---|---|---|---|---|
| Hill D, Spence J | 2015 | Transcriptional characterization of Human Lung Organoids (HLOs) using RNAseq | https://www.ebi.ac.uk/arrayexpress/experiments/E-MTAB-3339/ | Publicly available at EMBL ArrayExpress (E-MTAB-3339). |
| Finkbeiner S, Spence J | 2014 | Transcriptional Profiling of human pluripotent stem cells and and derived tissues | https://www.ebi.ac.uk/arrayexpress/experiments/E-MTAB-3158/ | Publicly available at EMBL ArrayExpress (E-MTAB-3158). |

The following previously published datasets were used:

| Author(s) | Year | Dataset title | Dataset ID and/or URL | Database, license, and accessibility information |
|---|---|---|---|---|
| Stamatoyannopoulos J | 2010 | University of Washington Human Reference Epigenome Mapping Project | Roadmap Epigenomics Series; http://www.ncbi.nlm.nih.gov/geo/query/acc.cgi?acc=GSE18927 | Publicly available at NCBI Gene Expression Omnibus (GSE18927). |
| Hallstrom BM | 2013 | RNA-seq of coding RNA from tissue samples of 95 human individuals representing 27 different tissues in order to determine tissue-specificity of all protein-coding genes | https://www.ebi.ac.uk/arrayexpress/experiments/E-MTAB-1733/ | Publicly available at EMBL ArrayExpress (E-MTAB-1733). |

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
