## [Decision Letter]

Thank you for sending your work entitled “In vitro generation of human pluripotent stem cell derived lung organoids” for consideration at *eLife*. Your article has been favorably evaluated by Janet Rossant (Senior editor and Reviewing editor) and 2 reviewers.

The Reviewing editor and the reviewers discussed their comments before we reached this decision, and the Reviewing editor has assembled the following comments to help you prepare a revised submission.

In this study, the authors describe a systematically delineated 3D protocol to generate human lung organoids from human pluripotent stem cells. HLOs possess organized proximal airway structures with apparently multiple differentiated epithelial cell types, as well as smooth muscle cells. There was enthusiasm for the overall concept and the potential of the system to be used to model lung differentiation, homeostasis and disease in vitro. This work should be of interest to the readership of *eLife*. However, there were concerns raised about the lack of in depth characterization of the different cell types generated and the lack of careful quantification of the extent of ordered differentiation in the system. For the paper to be reconsidered, we would ask you to address the various specific comments below, but to focus on careful characterization of the different lung cell types generated by extensive use of confocal imaging of co-staining with appropriate antibodies, such as FOXJ1, beta-tubulin, CC10, SCGB1A1, SCGB3A2, P63, Podoplanin, HOPX1, and more markers of SM compartment. It would also be important to provide more evidence that the organoids really regenerate at least partially the ordered branching seen in the lung- such that you could define a bronchioalveolar duct junction (BADJ) with the proper smooth muscle configuration. Further definition of the cell types arising from the mesenchyme would also be useful. Are there any cartilage cells or other types of cells such as lipofibroblasts, myofibroblasts, etc.

In addition, some quantification of the proportions of different cell types produced and comparison with other published lung differentiation protocols is required. We recognize that functional assessment of the cell types produced is difficult but use of EM identify lamellar bodies in type 2 cells would help.

Specific comments:

Reviewer #1:

Figure 1: Foregut endoderm should be characterized by double positive for FOXA2+SOX2+. SOX2 single expression is not enough for anterior endoderm identification.

Figure 2: especially Figure 2—figure supplement 1. In order to ensure NKX2.1+ are endoderm derived lung progenitors, co-staining with a battery of markers including FOXA2, SOX9, ID2, FOXP2 etc. are needed for comprehensive characterization.

Figure 3: In the text, the authors claimed that most spheres co-expressed FOXA2, SOX2, and NKX2.1 protein. However, only single marker staining was shown. Double and triple staining is highly recommended.

Figure 3: the authors claimed that their lung organoids contains bipotent alveolar progenitor markers including NMYC, ID2, NKX2.1, SPC, and HOPX based on their q-PCR analysis. Those differentiated markers need to validated with double or triple co-staining.

In supplemental figures for Figure 3, the authors indicated that the majority of cells lose epithelial identity and become vimentin positive mesenchymal cells over time. It is also still possible that mesenchyme overgrew epithelium.

Figure 4: Most differentiation marker staining is of poor quality and hard to interpret. For example, acetylated tubulin staining for ciliated cell is within the cell, rather than extend outside of the cell like cilia (Figure 4). CCSP is dot-like (Figure 4). P63 expression is a general epithelial stem cell markers, therefore co-staining of P63 with NKX2.1, FOXA2 and SOX2 are needed to ensure they are airway epithelial stem cells (Figure 4).

Figure 5: Again, all markers need to be co-stained with NKX2.1 to ensure the lung progenitor identity. In both Figure 4 and Figure 5, the authors need to provide quantitative data concerning the efficiency of airway and alveolar differentiation in their organoids.

*Reviewer #2*:

1) HLOs have the ability to generate mature airway cells, but predominantly possess a bipotent alveolar progenitors. Does this protocol favors to proximal lung development? Does the length of time in 3D culture alter the ratio of proximal versus distal?

2) Increase in HH signaling leads to an increase in NKX2-1 expression in culture. Since HH signaling is received by mesenchymal cells, what is mediating its effect on NKX2-1?

3) Combined inhibition of endogenous FGF signaling with SU and activation of HH with SAG caused an additional increase in NKX2-1 expression (21 fold vs. 6.5 fold with SAG only). However, treating foregut cultures with SU along did not alter NKX2.1 expression. What is the explanation for the increase in NKX2-1 expression with the SU and HH combination?

4) What effect does HH has on SMC differentiation in culture?

5) Comparing experiments in Figure 2 versus 2D, it seems that NKX2-1 and PAX8 induction in these two experiments are quite different. Does the efficiency vary quite a bit between experiments?

6) For Figure 2, the right panel, NOG/SB inhibition in endoderm cultures does not seem to have led to induction of PAX8, as was concluded in Results.

---

## [Author Response]

*In this study, the authors describe a systematically delineated 3D protocol to generate human lung organoids from human pluripotent stem cells. HLOs possess organized proximal airway structures with apparently multiple differentiated epithelial cell types, as well as smooth muscle cells*.

*There was enthusiasm for the overall concept and the potential of the system to be used to model lung differentiation, homeostasis and disease in vitro. This work should be of interest to the readership of* eLife*. However, there were concerns raised about the lack of in depth characterization of the different cell types generated and the lack of careful quantification of the extent of ordered differentiation in the system. For the paper to be reconsidered, we would ask you to address the various specific comments below, but to focus on careful characterization of the different lung cell types generated by extensive use of confocal imaging of co-staining with appropriate antibodies, such as FOXJ1, beta-tubulin, CC10, SCGB1A1, SCGB3A2, P63, Podoplanin, HOPX1, and more markers of SM compartment.*

We agree with the reviewers that HLOs were not adequately described in quantitative detail. To address these concerns, we have now added a significant amount of new imaging, morphometric analysis and bioinformatics data to help paint a clearer picture of the HLO. Specifically, we have conducted the following:

1) Quantitated the % of HLOs that possess proximal airway-like structures (Figure 7).

2) Quantitated the % of area in HLOs that are comprised or either proximal airway-like structures or distal airway-like structures (Figure 7).

3) Quantitated the % of cells in HLOs that express NKX2.1, P63, FOXJ1, SPC, PDPN (Figure 7).

4) Serial slices from confocal Z-stack images co-labeled with: i) ECAD/SOX9/SOX2; ii) P63/ECAD; and iii) P63/aSMA (Figure 3—figure supplement 6 and Figure 4—figure supplement 1).

5) Importantly, through our analysis, we observed that staining for some cell types, including Club cells, was very rare. Coupled with our observation that some cells were reminiscent of fetal airway progenitors (SOX9+/SFTPC+), this led us to hypothesize that the HLOs were fetal in nature. In order to determine if was indeed the case, we used RNA sequencing to compare HLOs to human fetal lungs and human adult lungs. This transcriptome-wide comparison strongly suggests that HLOs are very similar to fetal lung (Figure 8).

*It would also be important to provide more evidence that the organoids really regenerate at least partially the ordered branching seen in the lung- such that you could define a bronchioalveolar duct junction (BADJ) with the proper smooth muscle configuration*.

Our new analysis suggests there is some organoid-to-organoid variation in terms of general structure/architecture. That is, greater than 80% of HLOs examined (n=48) had definitive proximal airway-like structures (Figure 7), and that these proximal airway-like structures accounted for ∼20% of the organoid cross-sectional area. The remaining 80% of the HLO was comprised of distal airway-like structures, where SFTPC+ and HOPX+ cells were found. Importantly, we also show that these distal-like airway structures had abundant myofibroblasts, fibroblasts and smooth muscle cells (Figure 5), but not cartilage. However, we did not find evidence that HLOs undergo branching morphogenesis or form proper BADJ structures.

Much like HLOs, intestinal organoids (Spence et al., Nature 2011) are poorly organized in vitro and only obtain higher order structure when transplanted in vivo (Watson et al., Nature Medicine, 2014). This higher order structure may result from a more complex growth environment that may include new cellular inputs (vasculature, innervation), circulating metabolites, or by changes in the mechanical forces placed on tissue transplanted under a kidney capsule. However, at this time, all of these possibilities are pure speculation. We have full intentions of determining if transplanting HLOs into the in vivo environment will enhance structural organization, however, due to the need to establish surgical procedures in my lab, and the time it may take for HLOs to structurally mature in vivo (>12 weeks), we feel that these experiments are outside the scope of the current work.

Therefore, in order to be as forthright as possible, we have added into our Discussion section that we do not find evidence of a BADJ or branching morphogenesis.

*Further definition of the cell types arising from the mesenchyme would also be useful. Are there any cartilage cells or other types of cells such as lipofibroblasts, myofibroblasts, etc*.

We have now included a new figure showing immunofluorescent staining to examine the mesenchymal compartment of HLOs (Figure 5). We also stained HLOs with Safranin O, which identifies cartilage, and find that HLOs do not possess cartilage, whereas iPSC-derived teratomas had abundant Safranin O-positive cartilage staining.

*In addition, some quantification of the proportions of different cell types produced and comparison with other published lung differentiation protocols is required*.

We agree that comparing our results to those of others is important. It is interesting that re-reviewing the literature led us to the realization that some groups making very important contributions to the field did not conduct such a detailed analysis of the proportion of cell types. In this light, we have included a section in the Discussion comparing our analysis to those of others whenever data was available.

*We recognize that functional assessment of the cell types produced is difficult but use of EM identify lamellar bodies in type 2 cells would help*.

We have now conducted TEM and find clear evidence of lamellar bodies that are both within cells, and that have been secreted into the extracellular space.

Specific comments:

Reviewer #1:

Figure 1*: Foregut endoderm should be characterized by double positive for FOXA2+SOX2+. SOX2 single expression is not enough for anterior endoderm identification*.

We now provide low-magnification (wide field of view) co-staining of FOXA2/SOX2 for Endoderm, Foregut Endoderm, and Foregut endoderm treated with SAG or SAG+SU5402. This staining is now included in Figure 2—figure supplement 1.

Figure 2*: especially*
Figure 2—figure supplement 1*. In order to ensure NKX2.1+ are endoderm derived lung progenitors, co-staining with a battery of markers including FOXA2, SOX9, ID2, FOXP2 etc. are needed for comprehensive characterization*.

We now include NKX2.1/FOXA2 and NKX2.1/SOX9 co-staining for each condition in Figure 1—figure supplement 1.

Figure 3*: In the text, the authors claimed that most spheres co-expressed FOXA2, SOX2, and NKX2.1 protein. However, only single marker staining was shown. Double and triple staining is highly recommended*.

We have now conducted this analysis and include the data in Figure 3—figure supplement 3.

Figure 3*: the authors claimed that their lung organoids contains bipotent alveolar progenitor markers including NMYC, ID2, NKX2.1, SPC, and HOPX based on their q-PCR analysis. Those differentiated markers need to validated with double or triple co-staining*.

We apologize for the unclear text. The point we were trying to get across is that high NMYC and ID2 mRNA levels are consistent with an early lung state. We have done co-staining for SFTPC/SOX9 and HOPX/SOX9. Double-positive cells also suggest these may be less differentiated. We have tried to clarify the text around this problem.

In addition, rather than using a few molecular markers and co-stained cells to speculate on the “maturation state” of HLOs, we now directly address this using RNA sequencing. By comparing whole transcriptome signatures of HLOs to fetal lung, adult lung, ES cells and definitive endoderm. This data is now presented in Figure 8. Based on global RNA expression levels, this analysis strongly suggests that HLOs are more similar to fetal/immature lung.

*In supplemental figures for*
Figure 3*, the authors indicated that the majority of cells lose epithelial identity and become vimentin positive mesenchymal cells over time. It is also still possible that mesenchyme overgrew epithelium*.

This point is correct and well taken. We have now revised the text to encompass this possibility.

Figure 4*: Most differentiation marker staining is of poor quality and hard to interpret. For example, acetylated tubulin staining for ciliated cell is within the cell, rather than extend outside of the cell like cilia (*Figure 4*)*.

Indeed, the reviewer is correct about the intracellular AcTUB staining, however, we do not believe this is due to poor quality of staining. Rather, as we state in the text, we believe that these FOXJ1+ cells are not forming proper multi-ciliated structures and may reflect the immature cellular state of HLOs. Our results from growing HLOs on acellular human lung matrix support this idea, since growth on the human matrix promotes proper multicilliated structures.

*CCSP is dot-like (*Figure 4*)*.

We find these CCSP+ cells only very rarely, and again, believe their sparcity and staining pattern are reflective of a less differentiated tissue.

*P63 expression is a general epithelial stem cell markers, therefore co-staining of P63 with NKX2.1, FOXA2 and SOX2 are needed to ensure they are airway epithelial stem cells (*Figure 4*)*.

Point well taken. Unfortunately, due to antibody specificity, we were unable to conduct NKX2.1/P63 co-staining. Instead, we have conducted staining on serial sections to show that cells in adjacent sections are staining positive for both markers. Additionally, we show that P63+ cells also express SOX2. This data is presented in Figure 4—figure supplement 2.

Figure 5*: Again, all markers need to be co-stained with NKX2.1 to ensure the lung progenitor identity. In both*
Figure 4
*and*
Figure 5*, the authors need to provide quantitative data concerning the efficiency of airway and alveolar differentiation in their organoids*.

We have now conducted co-staining and quantitation of the HLO. This data is presented in Figure 6, Figure 6—figure supplement 1 and in Figure 7. Again, due to antibody specificity limitations, some of this analysis was conducted on serial sections.

Reviewer #2:

*1) HLOs have the ability to generate mature airway cells, but predominantly possess a bipotent alveolar progenitors. Does this protocol favors to proximal lung development? Does the length of time in 3D culture alter the ratio of proximal versus distal*?

Given our new quantitative analysis of the HLOs and RNAseq comparisons to adult and fetal lung, we do not believe that our methods favor proximal lung development. Our current thought is that the relatively low abundance of mature alveolar cells may be reflective of the immature/fetal nature of the HLO. Moreover, it is possible that our culture conditions, which utilize high levels of FGF10, may promote this less differentiated cellular state, and our data suggests that distal differentiation may require prolonged time in culture, since SFTPC begins to increase after ∼65 days in culture, and continues to increase at 110 days in culture, the latest time we have examined.

We believe that the best experiment to test these possibilities is to transplant the HLOs into a recipient host (NSG mouse), and determine if they show enhanced cellular differentiation. As mentioned above, we are currently working toward establishing the techniques required to conduct these experiments in the Spence lab, but are unable to do so at this time.

*2) Increase in HH signaling leads to an increase in NKX2-1 expression in culture. Since HH signaling is received by mesenchymal cells, what is mediating its effect on NKX2-1*?

This is a very good question, and one that does not have a definitive answer at this time. It is clear from our experiment in Figure 2 that HH signaling is required for NKX2.1 expression in NOG/SB treated foregut endoderm. When we inhibit HH with a small molecule, Sant-2, NKX2.1 expression is reduced. However, we still do not understand the mechanism by which this happens. One strong possibility is that HH signaling enhances mesenchymal proliferation and/or survival in HLOs, thus increasing important reciprocal signaling events between the two populations. Indeed, immunostaining of NOG/SB/F/CH spheroids (Figure 1—figure supplement 3) and NOG/SB/F/CH/SAG spheroids (Figure 3—figure supplement 5) for the mesenchymal marker Vimentin (VIM) revealed that there may be more mesenchymal cells present in SAG treated spheroids. However, differences between these groups were not significant by QRTPCR (data not shown).

*3) Combined inhibition of endogenous FGF signaling with SU and activation of HH with SAG caused an additional increase in NKX2-1 expression (21 fold vs. 6.5 fold with SAG only). However, treating foregut cultures with SU along did not alter NKX2.1 expression. What is the explanation for the increase in NKX2-1 expression with the SU and HH combination*?

We believe that our data supports a paradigm where both factors in combination (SAG+SU) support robust expression of NKX2.1 in more cells when compared with addition of SAG alone (Figure 1, Figure 1—figure supplement 1). In this scenario, SAG alone results in robust “patches” of NKX2.1+ cells in the monolayer whereas SAG+SU promotes robust protein expression of NKX2.1 in majority of cells within a field.

This may be due to the fact that with SAG alone, cells still receive endogenous FGF signals, and have more lineage choices to make, whereas removing FGF narrows the number of decisions a cell has such that more commit to an NKX2.1+ lineage.

*4) What effect does HH has on SMC differentiation in culture*?

This is an interesting question given previously described roles for HH on smooth muscle differentiation. We conducted several experiments in an attempt to address this reviewer question, and unfortunately, we did not see definitive changes in smooth muscle differentiation when we manipulated HH signaling (data not shown). We believe that this is likely due to experimental design. For example, it is possible we did not use high enough doses or long enough exposures to see an effect. These experiments are still ongoing, however, at this time we only have negative data to report and have chosen not to comment in the revised manuscript.

*5) Comparing experiments in*
Figure 2
*versus 2D, it seems that NKX2-1 and PAX8 induction in these two experiments are quite different. Does the efficiency vary quite a bit between experiments*?

We believe that this variability arises from our “endoderm control” group, in which we add ActivinA for four days, followed by only basal media with no growth factors for an additional 12 days. In this setting, cells that have been induced to become endoderm but given no additional cues begin to undergo stochastic differentiation into multiple lineages, as we have previously shown (Spence et al., 2011). Thus, in these types of experiments, while the trends in gene expression are usually reproducible, the level of the changes vary somewhat from experiment to experiment, as the reviewer points out in 2C vs. 2D.

*6) For*
Figure 2*, the right panel, NOG/SB inhibition in endoderm cultures does not seem to have led to induction of PAX8, as was concluded in Results*.

This reviewer brings up a good point, and we have tried to clarify the text surrounding this result, and we offer the following explanation; For this experiment, all conditions were analyzed after the same amount of time in culture. For the endoderm control, endoderm was induced for 4 days, and then placed in basal culture media for the duration of the experiment (12 days).

Similarly, the Foregut condition received ActivinA for four days, followed by four days of NOG/SB, and followed by 8 days in basal media. And so on. The likely reason PAX8 is not highly expressed in this experiment is that its expression is not maintained over time in the absence of any growth factors. This is supported by Figure 3—figure supplement 3, where we generated foregut endoderm (NOG/SB) and then treated it with FGF2 for 8 days, and observed many PAX8+ cells within the monolayer.